# IN-CONTEXT MULTI-OBJECTIVE OPTIMIZATION

**Xinyu Zhang**[1,2], **Conor Hassan**[1,2], **Julien Martinelli**[1,2], **Daolang Huang**[1,2], **Samuel Kaski**[1,2,3]
[1]Department of Computer Science, Aalto University, Finland
[2]ELLIS Institute Finland
[3]Department of Computer Science, University of Manchester, UK
`xinyu.2.zhang@aalto.fi`

## ABSTRACT

Balancing competing objectives is omnipresent across disciplines, from drug design to autonomous systems. Multi-objective Bayesian optimization is a promising solution for such expensive, black-box problems: it fits probabilistic surrogates and selects new designs via an acquisition function that balances exploration and exploitation. In practice, it requires tailored choices of surrogate and acquisition that rarely transfer to the next problem, is myopic when multi-step planning is often required, and adds refitting overhead, particularly in parallel or time-sensitive loops. We present TAMO, a fully amortized, universal policy for multi-objective black-box optimization. TAMO uses a transformer architecture that operates across varying input and objective dimensions, enabling pretraining on diverse corpora and transfer to new problems without retraining: at test time, the pretrained model proposes the next design with a single forward pass. We pretrain the policy with reinforcement learning to maximize cumulative hypervolume improvement over full trajectories, conditioning on the entire query history to approximate the Pareto frontier. Across synthetic benchmarks and real tasks, TAMO produces fast proposals, reducing proposal time by 50–1000× versus alternatives while matching or improving Pareto quality under tight evaluation budgets. These results show that transformers can perform multi-objective optimization entirely in-context, eliminating per-task surrogate fitting and acquisition engineering, and open a path to foundation-style, plug-and-play optimizers for scientific discovery workflows.

## 1 INTRODUCTION

Multi-objective optimization (MOO; Deb et al., 2016; Gunantara, 2018) is ubiquitous in science and engineering: practitioners routinely balance accuracy vs. cost in experimental design (Schoepfer et al., 2024), latency vs. quality in adaptive streaming controllers (Peroni & Gorinsky, 2025), or efficacy vs. toxicity in drug discovery (Fromer & Coley, 2023; Lai et al., 2025). In these settings, each evaluation of the black-box objectives can be slow or costly, making sample efficiency paramount; the goal is to obtain high-quality approximations of the Pareto front with a minimal number of queries.

The standard sample-efficient paradigm for such problems is Multi-objective Bayesian optimization (MOBO; Garnett, 2023): fit probabilistic surrogates for each objective, typically using Gaussian processes (GPs; Rasmussen & Williams, 2006), then select the next query by maximizing an acquisition that balances exploration–exploitation to efficiently improve a chosen multi-objective utility, such as hypervolume, scalarizations, or preference-based criteria (Daulton et al., 2020; Belakaria et al., 2019; Daulton et al., 2023b). While effective, this recipe has three drawbacks in real-world use. First, each new problem requires training surrogates from scratch and repeatedly optimizing the acquisition, adding non-trivial GP overhead that can bottleneck decision latency in parallel or time-sensitive settings. Second, performance critically depends on modeling choices (kernel, likelihood, acquisition, initialization), especially when data are scarce, a setting MOBO is intended to handle. Third, most acquisitions are myopic, optimizing a one-step gain, which can be suboptimal when Pareto-front discovery requires multi-step planning.

Amortized optimization (Finn et al., 2017; Amos et al., 2023) addresses these issues by shifting computation offline. The idea is to pre-train on a distribution of related optimization tasks, either generated synthetically or drawn from real, previously solved datasets. At test time, proposing a new

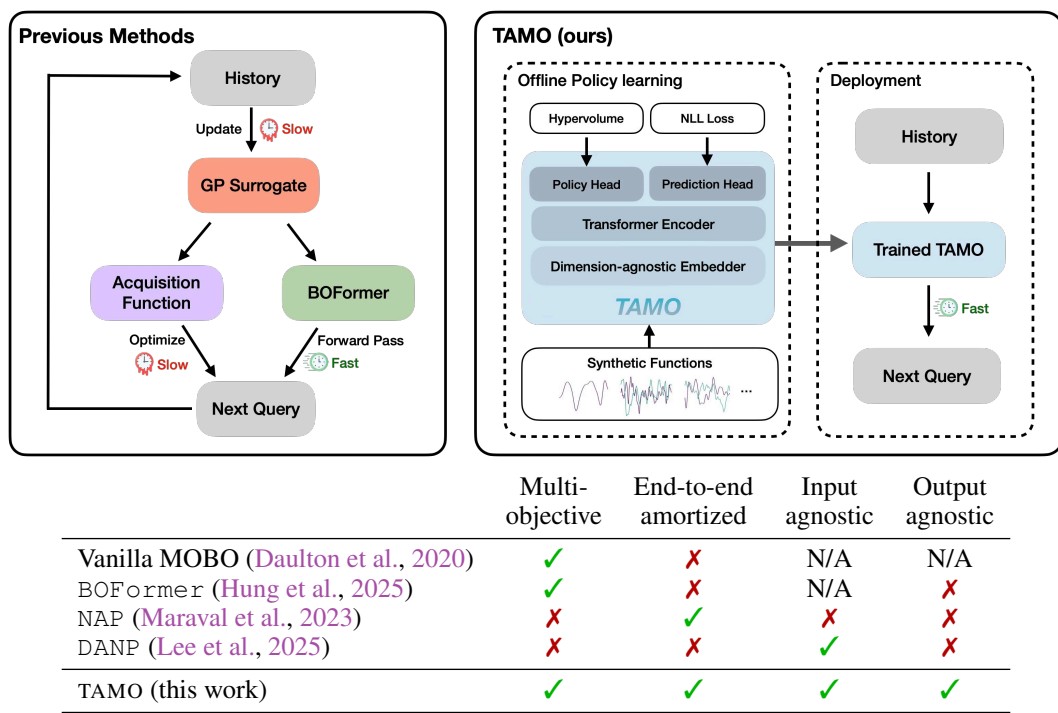

| | Multi-objective | End-to-end amortized | Input agnostic | Output agnostic |
|---|:---:|:---:|:---:|:---:|
| Vanilla MOBO (Daulton et al., 2020) | ✓ | ✗ | N/A | N/A |
| BOFormer (Hung et al., 2025) | ✓ | ✗ | N/A | ✗ |
| NAP (Maraval et al., 2023) | ✗ | ✓ | ✗ | ✗ |
| DANP (Lee et al., 2025) | ✗ | ✗ | ✓ | ✗ |
| TAMO (this work) | ✓ | ✓ | ✓ | ✓ |

Figure 1: **Comparison of multi-objective optimization workflows.** (Top left) Previous methods like traditional MOBO or acquisition-only amortized BOFormer (Hung et al., 2025) are bottlenecked by a slow process of fitting a GP surrogate. (Top right) TAMO is *fully amortized*: a dimension-agnostic transformer policy is trained once, offline, on diverse synthetic tasks, and at deployment maps the history to the next query in a single forward pass.

design then reduces to a single forward pass. Recent efforts have explored methods for amortized Bayesian optimization (Volpp et al., 2020; Chen et al., 2022; Maraval et al., 2023; Zhang et al., 2025; Hung et al., 2025), but few address the multi-objective setting. For instance, Hung et al. (2025) only amortizes the acquisition function calculation while still relying on a GP surrogate, and its pretrained model is tied to a fixed number of objectives, which prevents transfer across heterogeneous tasks. A method that tackles these challenges would let practitioners pool heterogeneous legacy datasets for pretraining, resulting in improved outcomes in scarce-data regimes. It would also enable reusing a single optimizer as design spaces and objective counts change, and issue instant proposals in closed-loop laboratories, high-throughput campaigns, reducing overhead when evaluations are cheap or parallel.

**Contributions.**

- We introduce TAMO, a fully amortized policy for multi-objective optimization that maps the observed history directly to the next query (Figure 1). Training uses reinforcement learning to optimize a hypervolume-oriented utility over entire trajectories, encouraging long-horizon rather than one-step gains. At inference, proposals are produced by a single forward pass.

- TAMO is dimension agnostic on both inputs and outputs: we introduce a transformer architecture with a novel dimension-aggregating embedder that jointly represents all input features and objective values regardless of dimensionality. This enables pretraining on heterogeneous tasks, synthetic or drawn from real meta-datasets, and transfer to new problems without retraining. To our knowledge, this is the first end-to-end, dimension-agnostic architecture for black-box optimization, let alone MOO (Figure 1, bottom).

- We evaluate TAMO on synthetic and real multi-objective tasks, observing $50\times$–$1000\times$ lower wall-clock proposal time than GP-based MOBO and baselines such as BOFORMER, which amortizes the acquisition but still relies on task-specific surrogates, while matching Pareto quality and sample efficiency. We further provide an empirical assessment of the generalization capabilities of TAMO, along with its sensitivity to deployment knobs.

## 2 PRELIMINARIES

**Multi-objective Optimization.** Consider a multi-objective optimization problem in which one aims to optimize a function $\boldsymbol{f}(\boldsymbol{x}) = [f_1(\boldsymbol{x}), \ldots, f_{d_y}(\boldsymbol{x})] \in \mathbb{R}^{d_y}$, and observations $\boldsymbol{y} = \boldsymbol{f}(\boldsymbol{x}) + \boldsymbol{\varepsilon}$ where $\mathcal{X} \subset \mathbb{R}^{d_x}$ is a compact search space. In many practical scenarios, it is not possible to find a single design $\boldsymbol{x}$ that is optimal for all objectives simultaneously. Instead, the notion of *Pareto dominance* is used to compare objective vectors. An objective vector $\boldsymbol{f}(\boldsymbol{x})$ *Pareto-dominates* another vector $\boldsymbol{f}(\boldsymbol{x}')$, denoted $\boldsymbol{f}(\boldsymbol{x}) \succ \boldsymbol{f}(\boldsymbol{x}')$, if $f^{(m)}(\boldsymbol{x}) \geq f^{(m)}(\boldsymbol{x}') \quad \forall m \in \{1, \ldots, d_y\}$ and there exists at least one objective $m'$ such that $f^{(m')}(\boldsymbol{x}) > f^{(m')}(\boldsymbol{x}')$. The *Pareto frontier* (PF) associated with a set of designs $X \subseteq \mathcal{X}$ is $\mathcal{P}(X) = \{\boldsymbol{f}(\boldsymbol{x}) : \boldsymbol{x} \in X, \nexists \boldsymbol{x}' \in X \text{ s.t. } \boldsymbol{f}(\boldsymbol{x}') \succ \boldsymbol{f}(\boldsymbol{x})\}$. A common goal in multi-objective optimization is to approximate the global frontier $\mathcal{P}(\mathcal{X})$ within a limited budget of $T$ function evaluations. One popular way to assess solution quality is the *hypervolume* (HV) indicator. For a reference point $\boldsymbol{r} \in \mathbb{R}^{d_y}$, the hypervolume $\text{HV}(\mathcal{P}(X) \mid \boldsymbol{r})$ measures how much of the objective space between $\boldsymbol{r}$ and the frontier $\mathcal{P}(X)$ is *"covered"* by Pareto-optimal points. In practice, the choice of $\boldsymbol{r}$ depends on domain-specific considerations (Yang et al., 2019).

**Reinforcement Learning.** Recent work leverages RL to learn *non-myopic* strategies in black-box optimization, accounting for downstream impact of each evaluation (Maraval et al., 2023; Zhang et al., 2025; Hung et al., 2025). An RL problem is a Markov decision process (MDP) (Sutton et al., 1998) with states, actions, transition dynamics, a reward encoding the optimization goal, and a discount factor weighting future vs. immediate rewards. The output is a policy $\pi_\theta(\boldsymbol{a} \mid \boldsymbol{s})$, a distribution over actions $\boldsymbol{a}$ for a state $\boldsymbol{s}$ that maximizes expected discounted return. In the *amortized* regime, one policy is trained offline on a distribution of tasks and then deployed across new problems without retraining.

## 3 TASK-AGNOSTIC AMORTIZED MULTI-OBJECTIVE OPTIMIZATION (TAMO)

We introduce TAMO, a fully amortized framework for multi-objective black-box optimization. TAMO encodes the optimization history and a candidate set with a transformer backbone and directly outputs acquisition utilities. To stabilize policy learning, we additionally incorporate a prediction task into our objective function. Section 3.1 details the construction of pretraining tasks for policy learning and prediction; Section 3.2 presents the TAMO architecture; Section 3.3 formalizes the RL objective and MDP; and Section 3.4 outlines training and inference procedures. Figure 1 illustrates our workflow.

### 3.1 PRETRAINING DATASET CONSTRUCTION

We pre-train TAMO on a diverse distribution of synthetic multi-objective optimization tasks, denoted by $p(\tau)$. Each task $\tau \sim p(\tau)$ is defined by a black-box function $\boldsymbol{f}_\tau : \mathcal{X} \subset \mathbb{R}^{d_x^\tau} \to \mathbb{R}^{d_y^\tau}$, where the input and output dimensions, $d_x^\tau$ and $d_y^\tau$, vary across tasks. This heterogeneity is key to learning a universal, dimension-agnostic policy. The full generative process for $p(\tau)$, which is based on GP priors with varied kernels and properties, is detailed in Appendix A.

During each training step, we sample two distinct mini-batches from task distributions to jointly optimize the model for policy learning and auxiliary prediction:

- **Policy-learning batches.** To train the decision-making policy, each batch contains a *history set* $\mathcal{D}^h = \{(\boldsymbol{x}^h, \boldsymbol{y}^h)\}_{h=1}^{N_h}$ and a *query set* $\mathcal{D}^q = \{\boldsymbol{x}^q\}_{q=1}^{N_q}$. The policy conditions on the history to select the most promising query from the query set.

- **Prediction batches.** To facilitate policy learning, we include an auxiliary prediction task to help the model learn the function landscape. We sample $N$ input-output pairs from a fresh function draw. These pairs are then randomly partitioned into a *context set* $\mathcal{D}^c = \{(\boldsymbol{x}^c, \boldsymbol{y}^c)\}_{c=1}^{N_c}$ and a *target set* $\mathcal{D}^p = \{\boldsymbol{x}^p\}_{p=1}^{N_p}$, on which the model performs an in-context regression task.

For each dataset type $s \in \{h, q, c, p\}$, the individual elements are denoted as $x_{t,j}^s$ where $t \in \{1, \ldots, T\}$ indexes the data point and $j \in \{1, \ldots, d_x^\tau\}$ indexes the input dimension. Similarly, for outputs we have $y_{t,k}^s$ where $k \in \{1, \ldots, d_y^\tau\}$ indexes the output dimension.

## 3.2 MODEL ARCHITECTURE

TAMO's architecture is designed around a single, shared backbone that operates in two distinct tasks during training: the *prediction task* and the *optimization task*. Each forward pass processes one mini-batch, either a prediction batch $(\mathcal{D}^{(c)}, \mathcal{D}^{(t)})$ or an optimization batch $(\mathcal{D}^{(h)}, \mathcal{D}^{(q)})$. While the input data types differ conceptually (context $\leftrightarrow$ history and target $\leftrightarrow$ query), they are processed by the same core components. The architecture comprises four parts: (i) a *dimension-agnostic embedder* mapping an observation to a vector regardless of input/output dimension; (ii) a *transformer encoder* that aggregates variable-size histories/contexts and exposes a compact summary; (iii) lightweight *task conditioning* via a small number of tokens; and (iv) two *heads*: a prediction head and a policy head. The dimension-agnostic embedder and the transformer encoder blocks are shared across tasks.

**(I) Dimension-agnostic embedder.** We apply learnable scalar-to-vector maps $e_x : \mathbb{R} \to \mathbb{R}^{d_e}$ and $e_y : \mathbb{R} \to \mathbb{R}^{d_e}$ dimension-wise, resulting in $\boldsymbol{e}_x = e_x(\boldsymbol{x}) \in \mathbb{R}^{d_x^\tau \times d_e}$ and $\boldsymbol{e}_y = e_y(\boldsymbol{y}) \in \mathbb{R}^{d_y^\tau \times d_e}$. Both functions $e_x$ and $e_y$ are parameterized as feedforward neural networks. After $L$ transformer layers on the concatenated tokens $[\boldsymbol{e}_x; \boldsymbol{e}_y]$, we apply learnable dimension-specific positional tokens $\boldsymbol{p}_x \in \mathbb{R}^{d_x^\tau \times d_e}$ and $\boldsymbol{p}_y \in \mathbb{R}^{d_y^\tau \times d_e}$ element-wise and mean-pool across the $d_x^\tau + d_y^\tau$ token axis to obtain a single representation $\boldsymbol{E} \in \mathbb{R}^{d_e}$. These positional tokens are randomly sampled for each batch from fixed pools of learned embeddings. We introduce the positional tokens to prevent the spurious symmetries over dimensionalities from a permutation-invariant set encoder, allowing the model to distinguish between features and objectives with the same values. During training, the embedder is applied to $\mathcal{D}^h$ and $\mathcal{D}^q$ to yield $\boldsymbol{E}^h$ and $\boldsymbol{E}^q$ for the *optimization* task, and to $\mathcal{D}^c$ and $\mathcal{D}^p$ to yield $\boldsymbol{E}^c$ and $\boldsymbol{E}^p$ for the *prediction* task. Each observation contributes $\mathcal{O}(1)$ tokens, so the cost scales with the number of observations, not with $d_x^\tau$ or $d_y^\tau$. Figure 2 summarizes the embedder.

**(II) Transformer encoder–decoder.** We stack $B := B_1 + B_2$ transformer layers and split them into two phases. For the ***first*** $B_1$ ***layers***, the *observed* tokens interact. The history (or context) tokens undergo self-attention to produce $\hat{\boldsymbol{E}}^h$ (or $\hat{\boldsymbol{E}}^c$), capturing intra-set structure. The query (or target) tokens then use cross-attention with the keys/values provided by $\hat{\boldsymbol{E}}^h$ (or $\hat{\boldsymbol{E}}^c$), yielding $\hat{\boldsymbol{E}}^q$ (or $\hat{\boldsymbol{E}}^p$). No task-specific tokens are present in $B_1$. This phase is the *only* path through which queries/targets access information from the history/context. Then, for the ***last*** $B_2$ ***layers***, the sequence is reduced to *only* the query/target tokens together with a small set of task-specific tokens (defined below). All history/context tokens are removed from the sequence. An attention mask enforces that, in these final layers, query/target tokens are permitted to attend *only* to the task-specific tokens (no query–query attention and no access to history/context). The task-specific tokens may self-attend among themselves.

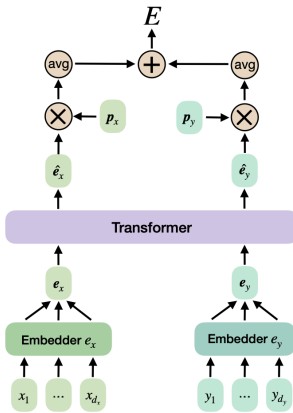

Figure 2: Dimension-agnostic embedder for a single observation.

**(III) Task-specific tokens.** Task-specific tokens are *introduced only at the entrance to the last $B_2$ layers*. For each task type, we introduce distinct tokens that guide the decoder's computation. For **prediction tasks**, the additional tokens comprise a *prediction* task token and the *output-index* positional token $\boldsymbol{p}_y^{(k)}$ indicating which scalar $y_{i,k}^p$ is to be predicted. For **optimization tasks**, the additional tokens comprise an *optimization* task token, a time-budget token $\boldsymbol{g}_{\text{time}} = \text{MLP}_\theta((T-t)/T)$, and an aggregated *input-dimension* token $\sum_{j=1}^{d_x^\tau} \boldsymbol{p}_x^{(j)}$. An attention mask restricts query/target tokens in $B_2$ to attend only to these tokens; the history/context never appears in $B_2$. This design yields constant overhead in $d_x^\tau$ and $d_y^\tau$, and linear cost in token size.

**(IV) Heads.** The architecture terminates in two heads that share the backbone but produce different outputs depending on the task.

*Prediction head.* Given the prediction tokens $\{\hat{\boldsymbol{E}}_i^p\}_{i=1}^{N_p}$ and an output-index positional token $\boldsymbol{p}_y^{(k)}$, the model produces, for each prediction input $\boldsymbol{x}_i^p$, the parameters of a $K$-component univariate Gaussian mixture that models the scalar $y_{i,k}^p$. Concretely, an MLP applied to $\hat{\boldsymbol{E}}_i^p$ yields mixture weights $\{\phi_{i\ell}\}_{\ell=1}^K$, means $\{\mu_{i\ell}\}_{\ell=1}^K$, and positive scales $\{\sigma_{i\ell}\}_{\ell=1}^K$, with weights normalized by a softmax

($\sum_{\ell=1}^{K} \phi_{i\ell} = 1$) and scales made positive via a softplus transform. The resulting predictive density is:

$$p\Big(y_{i,k}^p \mid \boldsymbol{x}_i^p, \mathcal{D}^c\Big) = \sum_{\ell=1}^{K} \phi_{i\ell} \, \mathcal{N}\big(y_{i,k}^p; \, \mu_{i\ell}, \, \sigma_{i\ell}^2\big). \tag{1}$$

Prediction tasks use samples disjoint from optimization tasks to prevent reward leakage.

*Policy head.* Given the query tokens $\{\hat{\boldsymbol{E}}_i^q\}_{i=1}^{N_q}$, the model assigns each query $\boldsymbol{x}_i^q$ a scalar acquisition utility $\alpha_i := \mathrm{MLP}_\theta(\hat{\boldsymbol{E}}_i^q)$. These utilities are converted into a categorical policy over the query set using a softmax:

$$\pi_\theta\big(\boldsymbol{x}_i^q \mid t, T, \mathcal{H}_{1:t-1}\big) = \frac{\exp(\alpha_i)}{\sum_{r=1}^{N_q} \exp(\alpha_r)}. \tag{2}$$

During training we sample actions from $\pi_\theta$; at inference we act greedily by selecting the next query with the largest probability. A detailed visualization of the full architecture is provided in Figure S1.

## 3.3 POLICY LEARNING

We cast the dimension-agnostic optimization problem as a Markov decision process (MDP):

- **State.** At step $t$, the state is $s_t = (\mathcal{D}^h, \, t, \, T)$, where $\mathcal{D}^h$ is the current historical observations and $T$ is the total budget.

- **Action.** The action selects a candidate index $i^\star \in \{1, \ldots, N_q\}$ and sets $\boldsymbol{x}_t = \boldsymbol{x}_{i^\star}^q$.

- **Reward.** After querying the objectives and observing $\boldsymbol{y}_t$, we update the history $\mathcal{D}^h := \mathcal{D}^h \cup \{(\boldsymbol{x}_t, \boldsymbol{y}_t)\}$ and define the *normalized hypervolume level*:

$$r_t = \frac{\mathrm{HV}\big(\mathcal{P}(\mathcal{D}^h) \mid \boldsymbol{r}\big)}{\mathrm{HV}_\tau^\star}, \qquad \mathrm{HV}_\tau^\star := \mathrm{HV}\big(\mathcal{P}(\mathcal{X}) \mid \boldsymbol{r}\big), \quad \mathrm{HV}\big(\mathcal{P}(\varnothing) \mid \boldsymbol{r}\big) = 0.$$

Here $\mathrm{HV}_\tau^\star$ is the task-wise hypervolume of the optimal frontier with respect to the fixed reference point $\boldsymbol{r}$. We set $\boldsymbol{r}$ to the componentwise worst value, $\boldsymbol{r} = [\hat{y}^{(1)}, \ldots, \hat{y}^{(d_y^\tau)}]$ with $\hat{y}^{(k)} := \min_{\boldsymbol{x} \in \mathcal{X}} f_\tau^{(k)}(\boldsymbol{x})$, which makes the hypervolume well defined and the reward bounded in $[0, 1]$. This ratio measures the fraction of maximum achievable hypervolume already captured by the current Pareto approximation (larger is better), and the normalization provides scale invariance across heterogeneous tasks (Teoh et al., 2025).

The policy $\pi_\theta(\boldsymbol{x} \mid s)$ maximizes the expected discounted return:

$$J(\theta) = \mathbb{E}_{\tau \sim p(\tau)}\left[ \mathbb{E}_{\pi_\theta}\left[ \sum_{t=1}^{T} \gamma^{t-1} r_t \right] \right], \tag{3}$$

and we estimate gradients with REINFORCE (Williams, 1992):

$$\nabla_\theta J(\theta) \approx \mathbb{E}_{\tau, \pi_\theta}\left[ \sum_{t=1}^{T} \nabla_\theta \log \pi_\theta(\boldsymbol{x}_t \mid s_t) R_t \right]. \tag{4}$$

Where $R_t = \sum_{k=t}^{T} \gamma^{k-t} r_k$. Since training uses synthetic tasks, $r_t$ is known exactly.

## 3.4 TRAINING AND INFERENCE

**Training.** We train TAMO in two phases. First, we warm up the backbone on the prediction task by minimizing a negative log-likelihood over $(\mathcal{D}^c, \mathcal{D}^p)$, which encourages accurate in-context regression and useful representations:

$$\mathcal{L}^{(p)}(\theta) = -\mathbb{E}_{\tau \sim p(\tau)}\left[ \frac{1}{N_p \, d_y^\tau} \sum_{i=1}^{N_p} \sum_{k=1}^{d_y^\tau} \log p\big(y_{i,k}^p \mid \boldsymbol{x}_i^p, \mathcal{D}^c\big) \right]. \tag{5}$$

After warm-up we transition to the joint training phase, where we optimize the policy with the trajectory objective $J(\theta)$ (Eq. 3), aligning the learning signal with improvements in Pareto quality alongside the prediction objective. The overall objective combines both terms:

$$\mathcal{L}(\theta) = \lambda_p \mathcal{L}^{(p)}(\theta) + \mathcal{L}^{(\text{rl})}(\theta), \qquad \mathcal{L}^{(\text{rl})}(\theta) = -J(\theta), \tag{6}$$

and is optimized with REINFORCE (Eq. 4); the coefficient $\lambda_p > 0$ trades off prediction and policy signals. In all experiments, we fixed $\lambda_p = 1.0$. Specifically, the prediction loss $\mathcal{L}^{(p)}$ and RL loss $\mathcal{L}^{(\text{rl})}$ are calculated from two distinct forward passes through the model with different datasets, which are then summed for a single backward pass. Training on full trajectories directly rewards long-horizon improvements, while amortization enables learning from many tasks offline.

**Inference.** At deployment, TAMO iteratively approximates the Pareto frontier under a budget $T$. We initialize the history with a random observation $\mathcal{D}^h \leftarrow \{\boldsymbol{x}_0^h, y_0^h\}$ and set $t \leftarrow 1$. Each iteration scores the current candidate set $\mathcal{D}^q$ with a single forward pass and proposes:

$$\boldsymbol{x}_t = \underset{\boldsymbol{x}_i^q \in \mathcal{D}^q}{\arg\max} \, \pi_\theta\big(\boldsymbol{x}_i^q \,\big|\, t, T, \mathcal{D}^h\big). \tag{7}$$

The proposed query $\boldsymbol{x}_t$ is then evaluated, and the resulting observation is used to update the history. This process is iterated until the cumulative evaluation cost meets the budget. Detailed descriptions of the algorithms for training and inference are provided in Appendix B .

## 4 RELATED WORK

**Multi-objective Bayesian Optimization (MOBO).** This line of work builds on Bayesian Optimization (Garnett, 2023), leveraging a combination of a statistical surrogate and an acquisition function to seek high-quality approximations to the Pareto set under tight evaluation budgets. Three families are prominent. *scalarization* methods convert MOBO into single-objective subproblems (e.g., ParEGO, TS-TCH; Knowles, 2006; Paria et al., 2020), letting practitioners reuse mature BO tooling and sweep preferences in parallel. *Indicator-based* methods optimize hypervolume-oriented criteria such as EHVI or HVKG (Daulton et al., 2020; 2023a), directly aligning the acquisition with the final Pareto-quality metric. Lastly, *information-theoretic* methods (PESMO, MESMO, PFES; Hernández-Lobato et al., 2016; Belakaria et al., 2021; Suzuki et al., 2020) select points that maximize information gain about the Pareto set or frontier, offering a principled exploration strategy. These approaches are effective but hinge on a carefully tuned, task-specific surrogate–acquisition pairing that must be refit and re-optimized at each iteration, all while remaining largely myopic. We instead learn a fully amortized policy that reduces design proposal to a single neural-network forward pass, dramatically lowering inference latency.

**Amortization and meta-learning.** Amortization replaces per-task inference with a model trained offline to operate *in-context*, exemplified by prior-data fitted transformers that achieve strong in-context performance after pretraining on large, heterogeneous datasets (Hollmann et al., 2025; Qu et al., 2025). In parallel, Conditional Neural Processes and their transformer variants learn predictors that condition on a context set and generalize via a single forward pass (Garnelo et al., 2018; Kim et al., 2019; Nguyen & Grover, 2022; Chang et al., 2025), with recent work extending them to *dimension-agnostic* settings (Dutordoir et al., 2023; Lee et al., 2025). These works focus on amortizing *prediction*. More recently, several studies leverage in-context pretrained neural processes for sequential decision-making (Huang et al., 2024; Zhang et al., 2025; Huang et al., 2025); our approach falls into this line as well.

**Amortized black-box optimization.** Several approaches train neural networks to amortize black-box optimization directly, typically by mapping histories to proposals or by predicting acquisition values, with success under scalar observations (Volpp et al., 2020; Chen et al., 2022; Yang et al., 2023; Maraval et al., 2023; Song et al., 2024; Huang et al., 2024), and even binary feedback (Zhang et al., 2025). Complementary to surrogate/acquisition amortization, transfer-BO with Monte Carlo Tree Search learns the search space itself by building a data-driven hierarchy of promising subregions on source tasks and reusing it to warm-start a new target before adapting online (Wang et al., 2024). However, none of these methods is simultaneously end-to-end (no per-task surrogate or acquisition), natively multi-objective, and capable of cross-dimensional transfer. The closest is BOFormer (Hung

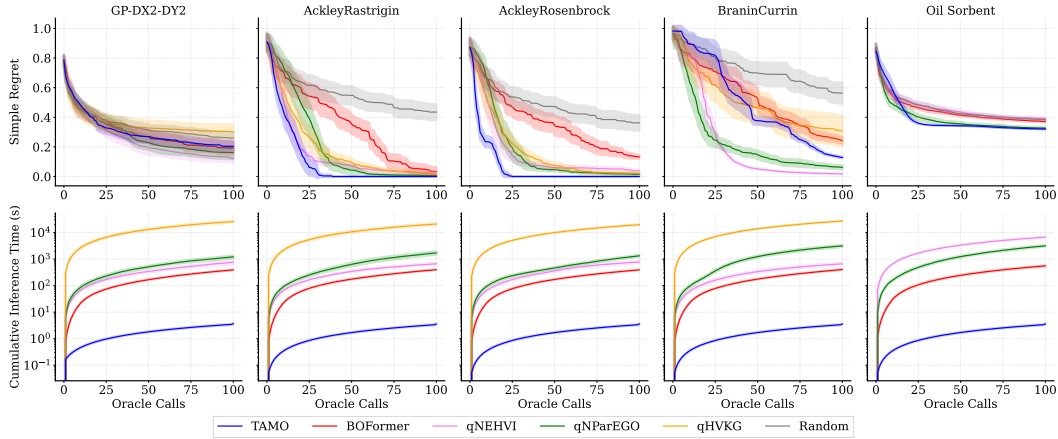

Figure 3: **Synthetic and real-world multi-objective benchmarks:** simple regret (top) and cumulative inference time (bottom) vs. oracle calls (mean $\pm$ 95% CIs over 30 runs). TAMO **achieves competitive regret while cutting proposal time by $50\times$–$1000\times$.**

et al., 2025), which uses sequence modeling to mitigate myopia in MOBO, but still relies on task-specific surrogates and fixed output dimensional setups, necessitating additional training when the dimension changes. We address all three by pretraining a fully amortized, foundation-style policy.

## 5  EXPERIMENTS

We evaluate TAMO on synthetic GP tasks and standard analytic testbeds, as well as on real-world problems (Section 5.1). Subsequently, Section 5.2 studies the generalization capabilities of TAMO, e.g., with respect to unseen task dimensionalities during training on both synthetic tasks and a real-world problem. We conclude with several ablation studies related to the batch size and query set size employed at inference time (Section 5.3). Additional experiments can be found in Appendix E. **We emphasize that a single pretrained model is used across all experiments.**

**Baselines.** We compare against strong MOBO baselines, including decomposition and indicator-based methods (qNParEGO Knowles 2006, qNEHVI Daulton et al. 2020, qHVKG Daulton et al. 2023a), sequence-modeling MOBO (BOFormer Hung et al. 2025), and a random search baseline. Baselines are tuned with their recommended defaults unless otherwise noted.

**Metrics.** We report performance via HV-based simple regret at a fixed evaluation budget. We also measure *wall-clock* proposal time end-to-end, which for GP-based baselines includes surrogate fitting and acquisition optimization, and for our method consists of a single forward pass. For single-objective, we additionally report standard simple regret.

**Implementation.** TAMO is implemented using PyTorch (Paszke et al., 2019). Hyperparameter settings can be found in Appendix D.3. Code is available on GitHub[1]. For all vanilla MOBO baselines, we used the implementation from the BoTorch library (Balandat et al., 2020). For BOFormer (Hung et al., 2025), we used the publicly available implementation and pretrained model from its official code repository. To ensure a fair comparison, the domain size (i.e., the size of the candidate query set) during testing is set to 2048, consistent with the configuration used for TAMO.

### 5.1  SYNTHETIC AND REAL-WORLD TASKS

**Synthetic examples.** On synthetic MOO testbeds (details in Section D.2), TAMO attains competitive or better simple regret across the entire budget (Figure 3). On GP-DX2–DY2, which is in-distribution for all methods (30 GP draws), TAMO performs on par with the best GP baselines. On the remaining three problems, *out-of-distribution* for all baselines, TAMO yields the strongest performance, except on Branin–Currin where qNEHVI and qNParEGO do better. We hypothesize this gap stems from the

---

[1]https://github.com/xinyuzc/in-context-moo

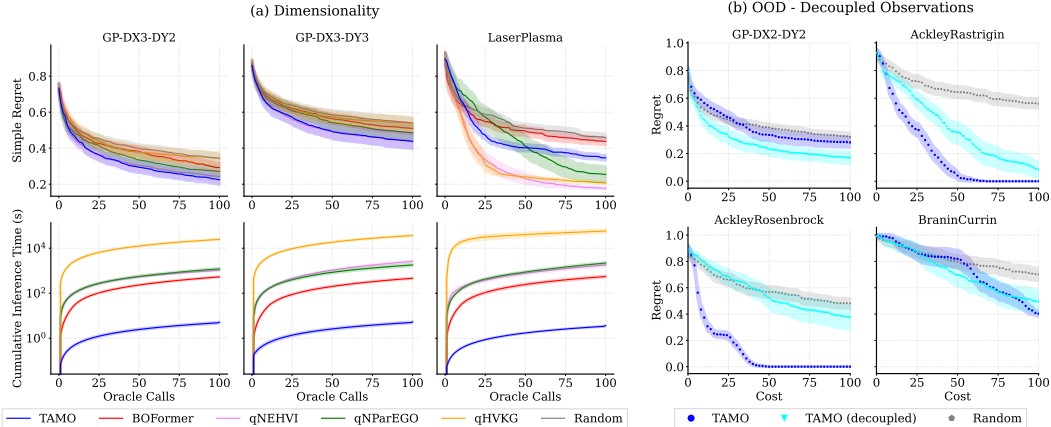

Figure 4: **Out-of-distribution evaluations. (a) Dimensionality:** simple regret (top) and cumulative inference time (bottom) on tasks whose input/output dimensions are unseen at pretraining. **(b) Decoupled observations:** regret vs. *cumulative cost* when, at step $t$, the optimizer may observe both objectives at cost 2 (dark blue) or only one at cost 1 (cyan). Curves show means with 95% confidence intervals over 60 runs (GP-DX3-DY2, GP-DX3-DY3) and 30 runs (others) with random initial observations. **TAMO shows reasonable generalization across unseen dimensionalities and decoupled feedback settings, while retaining orders-of-magnitude faster proposal times and broadly competitive regret.**

objectives in being well described by long length scales, outside the reach of our pretraining corpus: synthetic GP samples using lengthscales $\ell \sim \mathcal{N}(2/3, 0.5)$ over $[-5, 5]^{d_x}$ (Section D.3). Lastly, our method can also be applied effortlessly to single-objective BO, yielding competitive results compared to other GP-based alternatives (Figure S2).

**Real-world example**. We compare our model, pretrained only with synthetic GP samples, with other baselines on the real-world oil sorbent multi-objective problem (Daulton et al., 2022). The result is shown in Figure 3. TAMO remains competitive with GP-based alternatives, yielding the best performance, closely followed by qNParEGO.

**Wall-clock time.** Nevertheless, the primary advantage is speed: cumulative inference time is lower by roughly 50×–1000×, growing slowly with budget because each proposal is a single forward pass. By contrast, GP-based methods incur substantial overhead from repeated surrogate refits and acquisition optimization. Even `BOFormer`, which amortizes the acquisition but still relies on a GP surrogate, remains noticeably slower than TAMO.

## 5.2 Generalization

We investigate the generalization capabilities of TAMO in two different test-time scenarios: unseen dimensionalities, or decoupled observations.

**Out-of-distribution dimensionalities.** We test cross-dimensional transfer by pretraining TAMO on GP tasks with $d_x \in \{1, 2\}$ and $d_y \in \{1, 2, 3\}$, then evaluating on (i) GP-DX3–DY2 and GP-DX3–DY3, and (ii) the real-world LaserPlasma task ($d_x = 4$, $d_y = 3$; Section D.2), all with unseen input/output dimensionalities. On the synthetic OOD settings (Figure 4a, left, middle), TAMO attains regret comparable to the strongest GP baselines across the budget; even with 60 repetitions, we do not observe statistically decisive differences. On LaserPlasma (Figure 4a, right), TAMO improves over `BOFormer` (which amortizes only the acquisition) but trails conventional MOBO baselines in regret. Across all cases, TAMO retains orders-of-magnitude advantages in cumulative inference time.

**Decoupled observations.** We next test generalization to *decoupled* settings, where objectives can be measured independently, a common setting when jointly observing all objectives is infeasible or costly, also arising when historical logs contain partial objective labels. Budget $T = 100$ with cost 1 per objective: a full evaluation costs $d_y$, a single-objective probe costs 1. Hence, a coupled

policy can do at most $T/d_y$ full evals, while a *decoupled* one can take up to $T$ single-objective measurements. Figure 4b plots regret vs. cumulative cost. On GP-DX2–DY2, Ackley–Rastrigin, and Branin–Currin, the *decoupled* variant of TAMO closely tracks the coupled policy, indicating that TAMO can accommodate partial-feedback acquisition without retraining, offering a flexible trade-off between measurement cost and optimization progress. The exception is Ackley–Rosenbrock, where decoupling hurts performance, likely because the objectives peak at disparate locations, so single-objective measurements transfer poorly and bias the search toward one goal.

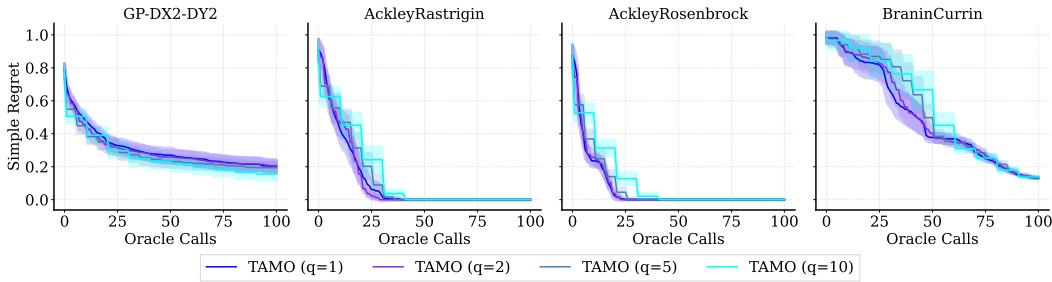

Figure 5: **Effect of batch size** on synthetic problems: simple regret for TAMO with $q \in \{1, 2, 5, 10\}$. Curves show means with 95% CIs over 30 runs. **Smaller $q$ converges fastest; larger $q$ incurs a mild slowdown, compatible with wall-clock savings for parallel evaluations.**

## 5.3 ABLATION STUDY

We examine TAMO's sensitivity to deployment knobs: *batch size* ($q$) and *query set size* ($N_q$), and quantify the accuracy–latency trade-offs they induce.

**Batch size.** We compare TAMO with $q \in \{1, 2, 5, 10\}$. For $q > 1$ we form batches via a light *fantasy* loop (Chang et al., 2022): pick $\boldsymbol{x}_t = \text{argmax}_{\boldsymbol{x}_i^q \in \mathcal{D}^q} \pi_\theta(\boldsymbol{x}_i^q \mid t, T, \mathcal{D}^h)$, predict a provisional outcome $\hat{\boldsymbol{y}}_t$, augment $\mathcal{D}^h \leftarrow \mathcal{D}^h \cup (\boldsymbol{x}_t, \hat{\boldsymbol{y}}_t)$, and repeat until $q$ points are chosen (i.e., $q$ forward passes). Across all problems, smaller batches reduce simple regret fastest; larger $q$ slows progress modestly, consistent with the lack of real feedback within a batch (hallucination). Nonetheless, degradation remains limited and all settings improve steadily with budget, indicating that when parallel evaluations are cheap, $q > 1$ can cut wall-clock time for a small accuracy cost (Figure 5).

**Query set size.** TAMO scores $N_q$ candidates per step; larger $N_q$ improves coverage but raises proposal cost. For $N_q \in \{256, 512, 1024, 2048\}$, regret is largely insensitive, except for Branin–Currin, where a small $N_q$ misses promising regions (Figure S3). Cumulative inference time grows roughly linearly with $N_q$. Even for $N_q = 2048$ (our default), TAMO remains much faster than other baselines.

## 6 DISCUSSION

We introduced TAMO, a fully amortized, task-agnostic policy for multi-objective black-box optimization. A single transformer backbone, trained offline with a prediction warm-up and a policy-level RL objective, maps histories to proposals in one forward pass and operates across varying input/output dimensionalities. Empirically, TAMO delivers proposal times 50 to 1000 times lower than conventional baselines while matching Pareto quality under tight evaluation budgets, sometimes even improving it.

**Limitations.** Our study highlights two main axes. First, pretraining data composition: although synthetic GP corpora provide scale and control, they may miss salient real-world structure. Downstream performance is likely sensitive to GP kernel families and smoothness, input metrics (e.g., Mahalanobis/rotated anisotropy), multi-output correlations (coregionalization models), observation models (homo-/heteroscedastic noise, decoupled observations), and simple landscape priors (e.g., adding a quadratic "bowl"). A systematic analysis that varies these ingredients would clarify how synthetic dataset design drives transfer. Second, inference currently assumes a discrete candidate pool, which can be restrictive in high-dimensional design spaces and in generative settings (e.g., *de novo* drug design) where the action space is continuous or combinatorial. Nevertheless, in pool-based

scenarios like high-throughput screening and library/catalog search, this assumption aligns with practice, and our method is highly effective.

**Perspectives.** We envision a bright future for TAMO as a *universal engine* for black-box optimization. The modular design invites extensions to black-box constraints, cost-aware and multi-fidelity settings, while retaining the single-pass interface. Beyond scalar observations, alternative data modalities, like preferential and multi-fidelity feedback, can also be incorporated with minor architecture changes. A key challenge is scaling to higher-dimensional spaces, with promising directions including factorizing the policy across input dimensions and moving from pool-based scoring to continuous policies or generative proposal mechanisms, in the spirit of amortized design networks (Foster et al., 2021). Efficient autoregressive mechanisms for transformer-based probabilistic models, such as the causal buffer of Hassan et al. (2026), could further accelerate sequential proposal generation by avoiding context re-encoding at each step. Lastly, as mentioned above, further work will investigate how the composition of the synthetic pre-training corpus influences downstream performance, an important direction for improving robustness and out-of-distribution behavior of amortized BO policies. Together, these advances position TAMO to serve as a foundation model-style optimizer that transfers across domains, objectives, and design spaces with minimal per-task tuning.

## ETHICS STATEMENT

All authors have read and will adhere to the ICLR Code of Ethics. This work does not involve human subjects, personally identifiable information, or sensitive attributes; experiments use synthetic data and standard public benchmarks. We are not aware of foreseeable harms from the methodology beyond typical risks of algorithmic misuse; the intended use is scientific and engineering optimization. Compute and environmental impact were kept reasonable (single-model pretraining and standard hardware); we report settings to support reproducibility. We will respect licenses of any third-party assets used and disclose any conflicts of interest if they arise.

## REPRODUCIBILITY STATEMENT

We document all experimental settings needed to facilitate replication: hyperparameters and optimizer details (Section D), procedures for pretraining dataset generation (Section A), and step-by-step algorithms for the training and prediction workflows (Section B). The code to reproduce our experiments is available at: https://github.com/xinyuzc/in-context-moo.

## ACKNOWLEDGEMENT

This work was supported by the Research Council of Finland (Flagship programme: Finnish Center for Artificial Intelligence FCAI). XZ and SK were supported by Research Council of Finland (RCF-NSF FinBioFAB, grant agreement 365982). CH and SK were supported by Business Finland (VirtualLab4Pharma, grant agreement 3597/31/2023) and the European Union (Horizon Europe, grant agreement 101214398, ELLIOT). DH and SK were supported by Research Council of Finland (Real-time AI-assistance with computationally rational user models, grant agreement 359207). SK was also supported by UKRI Turing AI World-Leading Researcher Fellowship (EP/W002973/1). The authors acknowledge the computational resources provided by the Aalto Science-IT Project from Computer Science IT.

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

# Appendix

The appendix is organized as follows:

- Section A describes the generative process leading to our synthetic pretraining dataset.

- Section B provides additional details regarding TAMO's workflow, with Algorithm S1 described pretraining.

- Section C provides a detailed architecture figure along with attention masks for TAMO.

- Section D provides further details regarding experiments, including model hyperparameters (Section D.3), computational resources (Section D.1) and test functions description (Section D.2).

- Section E contains additional experiments and analyses:
  - Single-objective Bayesian optimization
  - Ablation study on the query set size Effect of prediction warm-up and the prediction term weight $\lambda_p$ in the policy training loss
  - Ablation study on the discount factor $\gamma$ in the RL objective
  - Comparison between the standard multi-step and myopic TAMO variants
  - Effect of model size on optimization performance
  - Timing breakdown for GP baselines
  - Evaluation on real-world HPO-3DGS hyperparameter optimization tasks
  - Effect of pre-training dataset composition

- Section F displays several examples of GP samples used during pretraining (Section F.1), examples of mean prediction and proposed queries on GP samples (Section F.2, and Section F.3 in the decoupled setting).

- Section G describes to what extent Large Language Models (LLMs) were utilized throughout this work and manuscript.

## A    GENERATIVE PROCESS OF SYNTHETIC PRETRAINING DATASET

The model evaluated in Section 5 was trained on a dataset of GP draws. This dataset was constructed to include a variety of configurations, spanning diverse dimensionalities and function properties. All functions were generated inside $[-5.0, 5.0]^{d_x}$ using the following procedure:

- Input dimensionality $d_x \sim \mathcal{U}(\{1, 2\})$ and output dimensionality $d_y \sim \mathcal{U}(\{1, 2, 3\})$.

- Regarding output correlations, with probability $1/2$, either independent output dimensions are sampled, or they are drawn from a multi-task GP, with task covariance defined as $k(i, j) = (\boldsymbol{B}\boldsymbol{B}^T + \text{diag}(\boldsymbol{v}))_{i,j}, i, j \in \{1, \cdots, d_y\}$. In this case, $\boldsymbol{B}$ is a low-rank matrix with rank $r \sim \mathcal{U}(\{1, \cdots, d_y\})$.

- The data kernel along each output dimension is equally sampled from the RBF, Matérn-3/2, Matérn-5/2 kernels, with standard deviation $\sigma \sim U[0.1, 1.0]$ and lengthscale $l \sim \mathcal{N}(2/3, 0.5)$ truncated to the range $[0.1, 2.0]$.

- The sampled function values, $\boldsymbol{y}$, were centered and normalized to lie within $[-1, 1]^{d_y}$.

Examples with different dimensionalities are illustrated in Figures S11, S12, S13 and S14 (Section F.1).

## B    PRETRAINING AND TEST-TIME ALGORITHMS

Algorithm S1 and Algorithm S2 describe the pre-training loops and test-time optimization procedure, respectively.

---

**Algorithm S1** TAMO Pre-Training Algorithm

---

**Require:** task distribution $p(\tau)$, prediction context size $N_c$, prediction target size $N_p$, query budget $T$, number of burn-in iterations $\eta$, number of total iterations num_total_iterations
1: **for** iteration $i = 1, \ldots,$ num_total_iterations **do**
2:                                                                                                  $\triangleright$ Prediction task
3:     Sample a task $\tau \sim p(\tau)$
4:     Sample prediction batches $\mathcal{D}^c = \{(\boldsymbol{x}_i^c, y_i^c)\}_{i=1}^{N_c}$ and $\mathcal{D}^p = \{\boldsymbol{x}_i^p\}_{i=1}^{N_p}$ from $\tau$
5:     Model predicts outcomes: $p(y_{i,k}^p \mid \boldsymbol{x}_i^p, \mathcal{D}^c), \forall \boldsymbol{x}_i^p \in \mathcal{D}^p$
6:     **if** $i \leq \eta$ **then**
7:         Update model by minimizing the prediction loss $\mathcal{L}^{(p)}$ (Equation 5)
8:     **else**                                        $\triangleright$ Policy learning task after burn-in phase
9:         Sample a new task $\tau \sim p(\tau)$
10:         Sample query set $\mathcal{D}^q$
11:         Initialize a history set $\mathcal{D}^h \leftarrow \{(\boldsymbol{x}_0^h, y_0^h)\}, \boldsymbol{x}_0^h \sim \mathcal{D}^q$
12:         Set reference point $\boldsymbol{r}$ and calculate optimal Hypervolume: $\text{HV}^* \leftarrow \text{HV}(\mathcal{P}(\mathcal{X}) \mid \mathbf{r})$
13:         Initialize Pareto set $\mathcal{P} \leftarrow \{y_0^h\}$
14:         **for** $t = 1, \ldots, T$ **do**
15:             Select next query point: $\boldsymbol{x}_t \sim \pi_\theta(\cdot \mid \mathcal{D}^h, t, T)$
16:             $y_t \leftarrow \text{Evaluate}(\boldsymbol{x}_t, \tau)$
17:             Update history set: $\mathcal{D}^h \leftarrow \mathcal{D}^h \cup \{(\boldsymbol{x}_t, y_t)\}$
18:             Update Pareto set: $\mathcal{P} \leftarrow \mathcal{P} \cup \{y_t\}$
19:             Compute reward: $r_t = \frac{\text{HV}(\mathcal{P}|\mathbf{r})}{\text{HV}^*}$
20:         **end for**
21:         Update model using the overall objective $\mathcal{L}$ (Equation 6)
22:     **end if**
23: **end for**

---

**Algorithm S2** TAMO Test-Time Algorithm

---

**Require:** Pre-trained TAMO model, new task $\tau_{\text{test}}$, query budget $T$, initial history set $\mathcal{D}_0^h := \{\boldsymbol{x}^h, y^h\}$ (with random samples if empty),
1: $\mathcal{D}^h \leftarrow \mathcal{D}_0^h$                                                        $\triangleright$ Initialize the history set
2: $\mathcal{P} \leftarrow \{y^h\}$                                                          $\triangleright$ Initialize the Pareto set
3: **for** $t = 1, \ldots, T$ **do**
4:     $\mathbf{x}_t \sim \pi_\theta(\cdot \mid \mathcal{D}^h, t, T)$                           $\triangleright$ Sample the next query location
5:     $y_t \leftarrow \text{Evaluate}(\mathbf{x}_t, \tau_{\text{test}})$
6:     $\mathcal{D}^h \leftarrow \mathcal{D}^h \cup \{(\mathbf{x}_t, y_t)\}$                      $\triangleright$ Update the history set
7:     $\mathcal{P} \leftarrow \mathcal{P} \cup \{y_t\}$         $\triangleright$ Update the Pareto set with the new observation
8: **end for**
9: **return** $\mathcal{D}^h, \mathcal{P}$

---

# C    DETAILED MODEL ARCHITECTURE

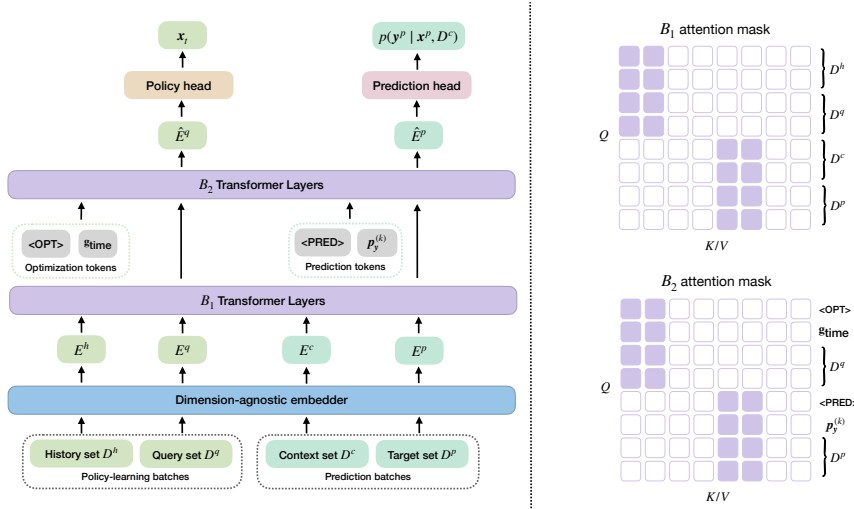

Figure S1: Detailed Architecture of TAMO.

# D    EXPERIMENTAL DETAILS

## D.1    COMPUTATIONAL RESOURCES

We trained TAMO on one NVIDIA H100 80GB HBM3 GPU. All models are evaluated on Tesla V100-SXM2-32GB GPUs.

## D.2    TEST FUNCTIONS

**GP samples optimization.**    This benchmark comprises 30 independent multi-output GP draws with $d_x = 2$ inputs and $d_y = 2$ objectives in the *dimensional in-distribution* setting (Section 5.1), and $d_x = 3$, $d_y \in \{2, 3\}$ in the *dimensional out-of-distribution* setting (Section 5.2). We sample each task using the same data-generating process described in Section D.3 and report average performance over the 30 draws.

**Ackley–Rastrigin** $d_x = 2, d_y = 2$.    Two-objective problem formed by pairing Ackley and Rastrigin and *maximizing* their negations:

$$\text{Ackley}(\boldsymbol{x}) = -20 \exp\Big( -0.2 \sqrt{\tfrac{1}{2} \sum_{i=1}^{2} x_i^2} \Big) - \exp\Big( \tfrac{1}{2} \sum_{i=1}^{2} \cos(2\pi x_i) \Big) + e + 20,$$

$$\text{Rastrigin}(\boldsymbol{x}) = 20 + \sum_{i=1}^{2} \big( x_i^2 - 10 \cos(2\pi x_i) \big),$$

And we set $f^{(1)}(\boldsymbol{x}) = -\text{Ackley}(\boldsymbol{x}), \ \ f^{(2)}(\boldsymbol{x}) = -\text{Rastrigin}(\boldsymbol{x})$.

**Ackley–Rosenbrock** $d_x = 2, d_y = 2$.    We pair Ackley (above) with Rosenbrock:

$$\text{Rosenbrock}(\boldsymbol{x}) = 100(x_2 - x_1^2)^2 + (1 - x_1)^2.$$

And we set $f^{(1)}(\boldsymbol{x}) = -\text{Ackley}(\boldsymbol{x}), \ \ f^{(2)}(\boldsymbol{x}) = -\text{Rosenbrock}(\boldsymbol{x})$.

**Branin–Currin** $d_x = 2, d_y = 2$. Branin:

$$\text{Branin}(x_1, x_2) = \left(x_2 - bx_1^2 + cx_1 - r\right)^2 + s\left(1 - t\right)\cos(x_1) + s,$$
$$\text{where } b = \tfrac{5.1}{4\pi^2}, \ c = \tfrac{5}{\pi}, \ r = 6, \ s = 10, \ t = \tfrac{1}{8\pi},$$

and Currin:

$$\text{Currin}(\boldsymbol{z}) = \left(1 - e^{-1/(2z_2)}\right)\frac{2300z_1^3 + 1900z_1^2 + 2092z_1 + 60}{100z_1^3 + 500z_1^2 + 4z_1 + 20}.$$

We maximize $f^{(1)}(\boldsymbol{x}) = -\text{Branin}(\boldsymbol{x})$ and $f^{(2)}(\boldsymbol{x}) = -\text{Currin}(\boldsymbol{x})$.

**Oil sorbent** $d_x = 2, d_y = 2$. We also evaluate on the oil-sorbent multi-objective problem (Wang et al., 2020; Daulton et al., 2022). The original task controls a material's composition and manufacturing with 5 ordinal and 2 continuous parameters to jointly maximize three objectives: `oil absorbing capacity`, `mechanical strength`, and `water contact angle`. In our study, we fix the ordinal parameters to constant values to obtain a 2D continuous design space with the same three objectives.

**Laser-Plasma** $d_x = 4, d_y = 3$. We evaluate on the laser–plasma acceleration dataset (Irshad et al., 2023), which contains 1025 particle-in-cell simulations of a laser wakefield accelerator. Each record provides 4 continuous inputs (`plasma density`, `upramp length`, `laser focus`, `downramp length`) and 3 objectives (`total charge`, `distance of median`, `target energy`). To obtain a continuous black-box from tabulated simulations, we perform linear interpolation. This task differs in dimensionality from our pretraining distribution, providing an OOD evaluation of cross-dimensional transfer.

**Normalization.** For all problems, we linearly rescale inputs to a common domain $[-5, 5]^{d_x}$ and rescale each objective independently to $[-1, 1]$ prior to logging and hypervolume computation.

### D.3 HYPERPARAMETERS

| Dimension-Agnostic Embedder | |
| --- | --- |
| Number of learnable positional tokens for $x$ | 4 |
| Number of learnable positional tokens for $y$ | 3 |
| Number of Transformer layers ($L$) | 4 |
| Dimension of $\mathbf{e}_x$ and $\mathbf{e}_y$ | 64 |
| **Transformer Encoder–Decoder** | |
| Dimension of Transformer inputs | 64 |
| Point-wise feed-forward dimension of Transformer | 256 |
| Number of self-attention layers in Transformer ($B$) | 8 |
| Number of self-attention heads in Transformer | 4 |
| **Heads** | |
| Number of hidden layers in policy head | 3 |
| Number of components in GMM head ($K$) | 20 |
| Number of hidden layers in MLP for each GMM component | 3 |
| **Training** | |
| Number of iterations | 400000 |
| Number of burn-in iterations | 393500 |
| Initial learning rate for warm-up iterations ($\mathrm{lr}_1$) | $1 \cdot 10^{-4}$ |
| Initial Learning rate after warm-up ($\mathrm{lr}_2$) | $4 \cdot 10^{-5}$ |
| Learning rate scheduling | Linearly increase from 0 to $\mathrm{lr}_1$ in the first $5\%$ of total iterations; Cosine decay to 0 over total iterations |
| Size of prediction batch | 32 |
| Size of policy-learning batch | 16 |
| Weight on prediction loss ($\lambda_{\mathrm{rl}}$) | 1.0 |
| discount factor ($\gamma$) | 0.99 |
| Size of context set | $N_c \sim U[2, 50 \cdot d_x^\tau]$ |
| Size of target set ($N_t$) | $300 - N_c$ |
| Size of query set ($N_q$) | 256 |
| Optimization budget $T$ | 100 |
| Noise level $\sigma$ | 0.0 |
| Number of initial observations during pretraining | 1 |
| **Evaluation** | |
| Number of initial observations during test time | 1 |
| Noise level $\sigma$ | 0.0 |
| Size of query set ($N_q$) | 2048 |
| Optimization budget ($T$) | 100 |

Table S1: Hyperparameter settings for TAMO evaluated in Section 5.

## E ADDITIONAL EXPERIMENTS

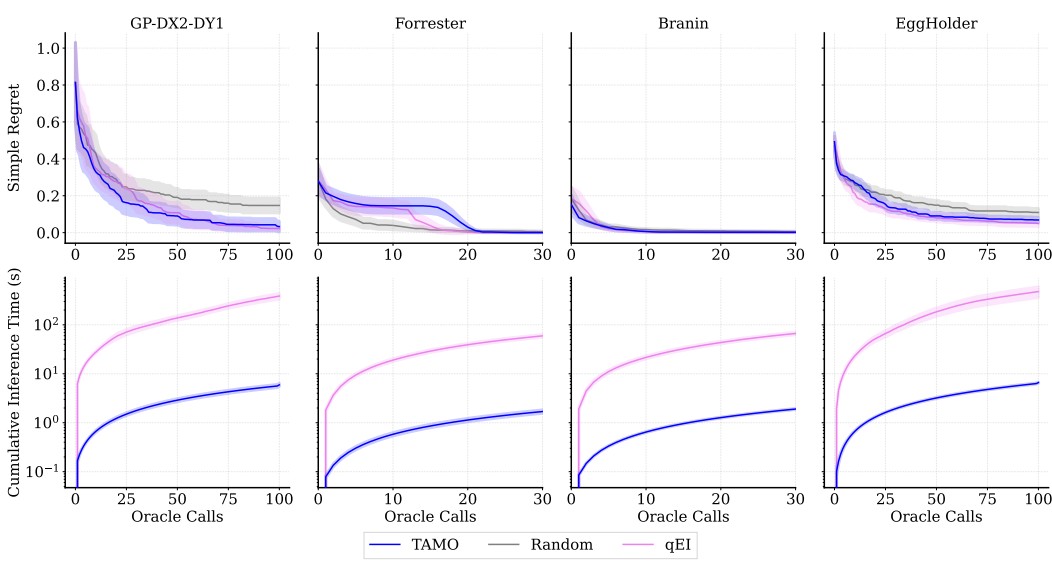

Figure S2: Simple regret and inference time on **synthetic examples for single-objective optimization**. Mean with $95\%$ confidence intervals computed across 30 runs with random initial observations. **Again, TAMO matches state-of-the-art regret while dramatically reducing proposal time.**

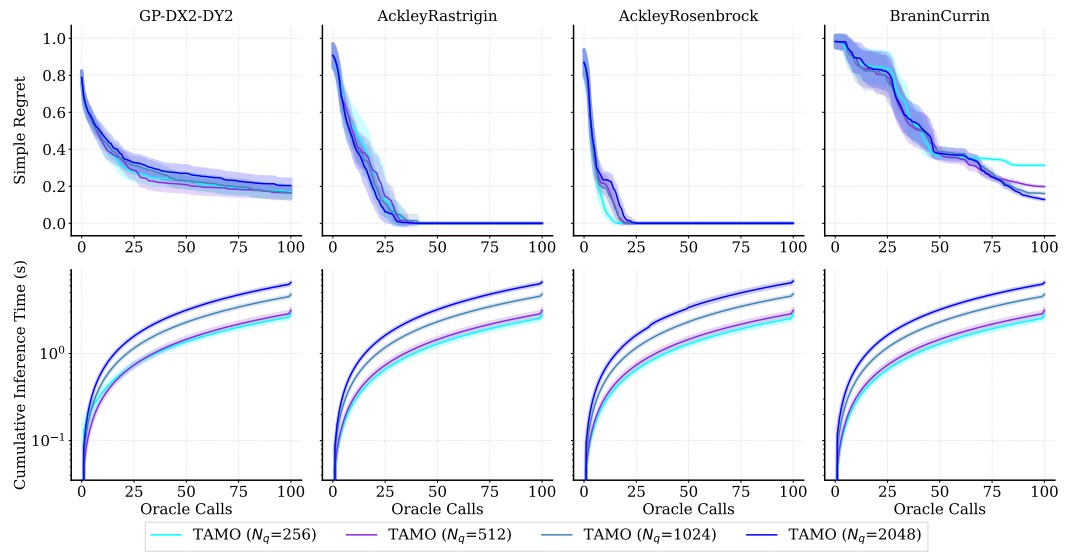

Figure S3: **Effect of query set size.** Simple regret (top) and cumulative inference time (bottom) for TAMO with $N_q \in \{256, 512, 1024, 2048\}$ on four synthetic tasks. Means with 95% CIs over 30 runs. **Larger $N_q$ increases wall-clock roughly linearly while leaving regret essentially unchanged.**

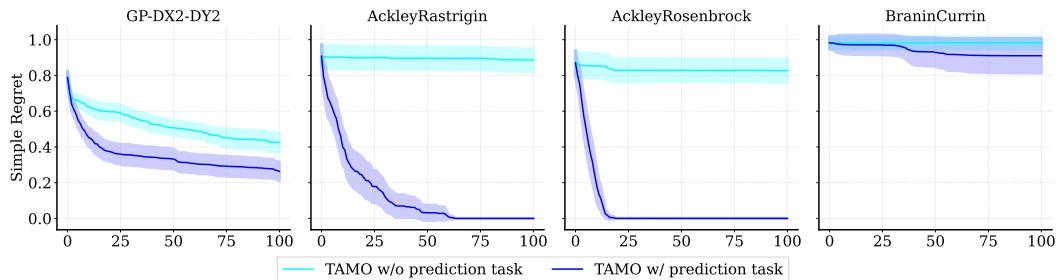

Figure S4: **Removing the prediction warm-up and prediction term from the training loss (Equation 6).** Simple regret on four synthetic tasks. Means with 95% CIs over 30 runs. **Introducing an auxiliary prediction task before and during policy training is decisive.**

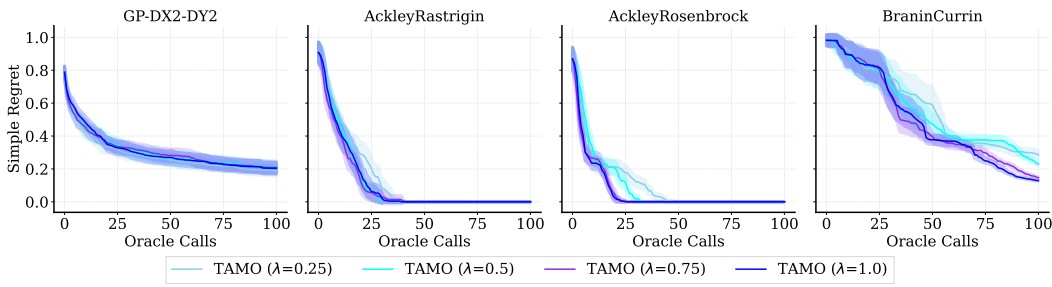

Figure S5: **Effect of the prediction term weight $\lambda_p$ in the training loss (Equation 6) during policy training.** Simple regret on four synthetic tasks. Means with 95% CIs over 30 runs. **Once policy training starts, performance is relatively insensitive to $\lambda_p$, with slightly better results for larger weights.**

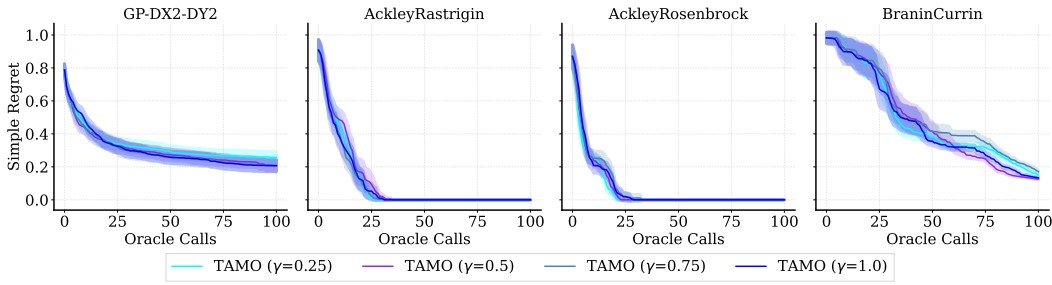

Figure S6: **Effect of the discount factor $\gamma$ in the RL objective (Equation 4) during policy training.** Simple regret on four synthetic tasks. Means with 95% CIs over 30 runs. **Overall, performance is fairly robust to $\gamma$.**

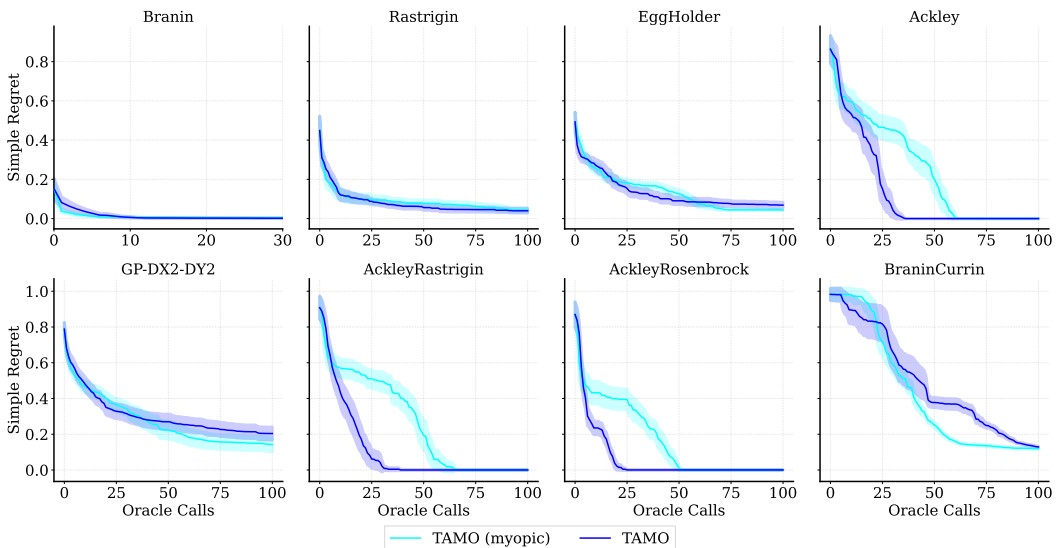

Figure S7: **Effect of the pretraining horizon in the RL objective (Equation 4) during policy training.** We compare a myopic variant of TAMO, pretrained with horizon $T = 1$, to the standard TAMO pretrained with $T = 100$. For the non-myopic model, each pretraining episode starts from a single randomly sampled context point, and the policy acts for up to 100 steps. For the myopic case, each episode starts from a randomly sampled context set, and the policy proposes a single additional point, so every decision is strictly one-step. Simple regret on four synthetic tasks for single objective optimization (top) and multi-objective optimization (bottom). Means with 95% CIs over 30 runs. **Results clearly advocate for longer pretraining horizons, except for the BraninCurrin problem.**

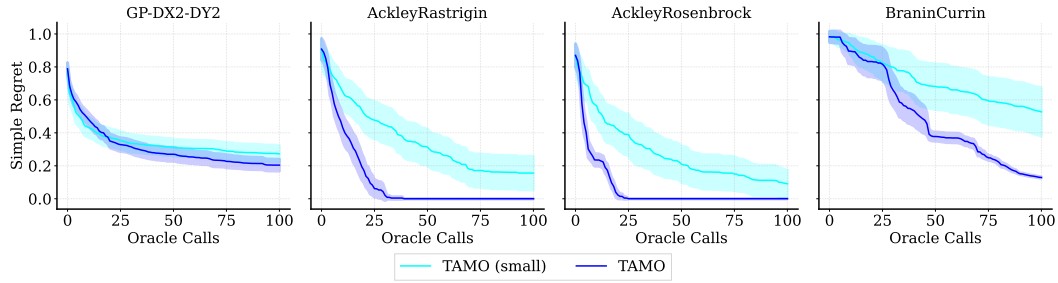

Figure S8: **Effect of model size on optimization performance.** We compare a smaller TAMO variant, trained for the same number of iterations as the standard model but using 2 Transformer layers per module (dimension-agnostic embedder, $B_1$ layers, $B_2$ layers) instead of 4. Simple regret on four synthetic tasks. Means with 95% CIs over 30 runs. **While the smaller model remains competitive, the larger backbone consistently attains lower regret, especially on the more challenging tasks.**

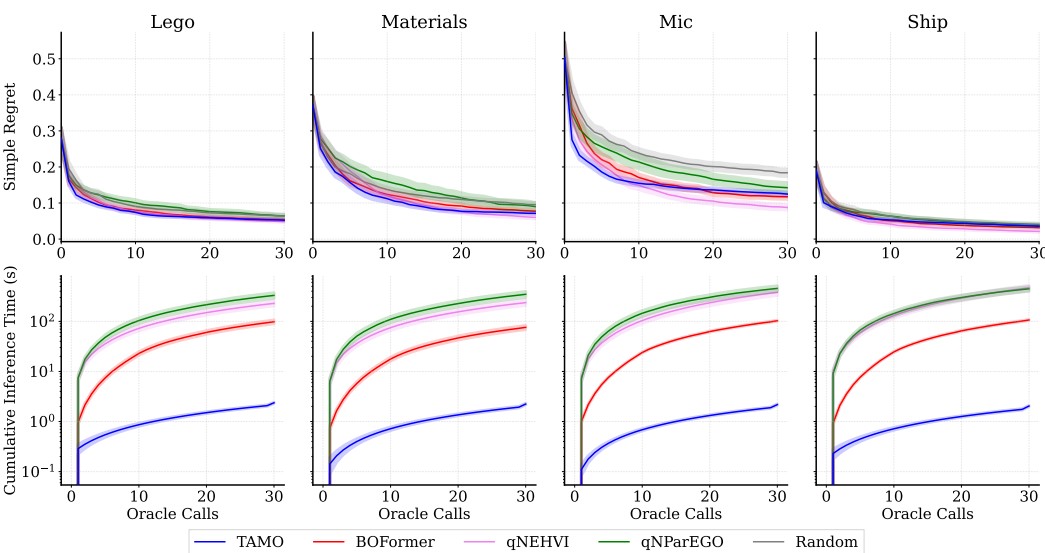

Figure S9: **Transfer to HPO-3DGS hyperparameter optimization tasks** Models are pretrained on synthetic GP data with $d_x = 5, d_y \in \{2, 3\}$, then evaluated on four 3D-Gaussian-Splatting scenes (Lego, Materials, Mic, Ship; $d_y = 2$ for Lego/Materials and $d_y = 3$ for Mic/Ship). Means with 95% CI over 50 runs. **Across all scenes, TAMO is competitive with the best-performing GP-based methods (with qNEHVI slightly ahead on Mic), while achieving substantially lower inference time.**

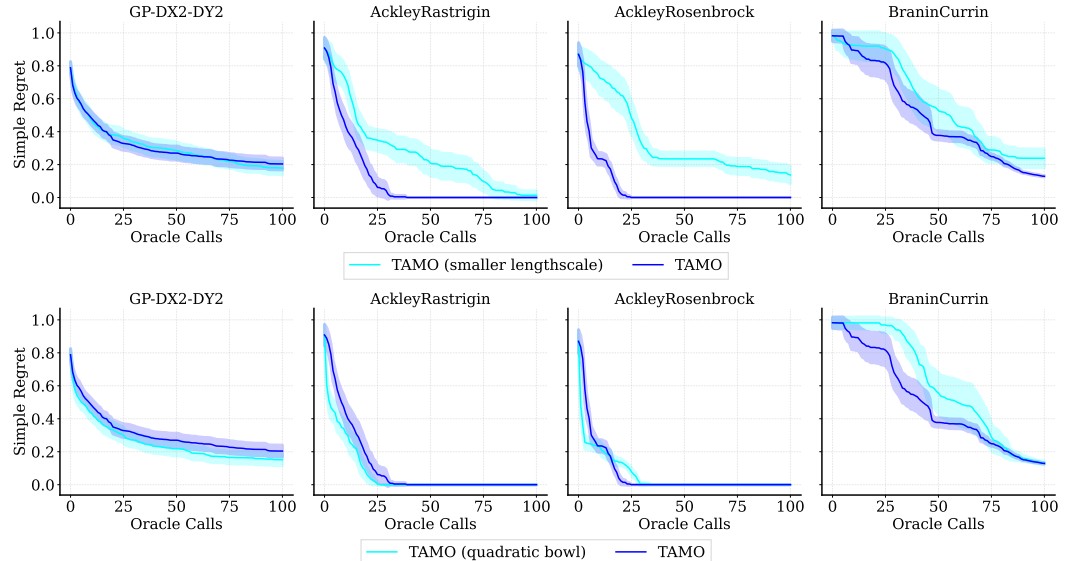

Figure S10: **Effect of pretraining-prior composition on optimization performance.** We compare the original TAMO with two variants trained on modified synthetic priors. **Top:** *Small-lengthscale* prior, where GP draws use shorter kernels by truncating the original lengthscale prior $\mathcal{N}(2/3, 0.5^2)$ from $[0.1, 2.0]$ to $[0.1, 0.5]$. **Bottom:** *Quadratic-bowl* prior, where for each of the $1 \le m \le d_y$ objectives we draw a GP and then sample an "optimum location" $\boldsymbol{x}^{\star(m)}$ uniformly in the design space, adding a quadratic term $\|\boldsymbol{x} - \boldsymbol{x}^{\star(m)}\|^2$ to the $m$-th objective. Simple regret on four synthetic benchmarks; curves show means with 95% confidence intervals over 30 runs. **The small-lengthscale prior degrades performance, while the quadratic-bowl prior maintains or improves performance on GP and Ackley-based tasks but hurts performance on Branin–Currin.**

# F  VISUALIZATION EXAMPLES

## F.1  EXAMPLES OF GP SAMPLES FROM PRETRAINING

Figures S11 to S14 show some examples of the GP samples used for pre-training.

## F.2  EXAMPLES OF INFERENCE

Figure S15 and Figure S16 show examples of mean predictions and proposed queries within a total budget $T = 100$ on GP samples, for input dimension $d_x = 2$ and output dimensions $d_y = 1$ and $d_y = 2$, respectively.

## F.3  EXAMPLES OF DECOUPLED OBSERVATIONS

Figure S17 shows examples of mean predictions and proposed queries under the decoupled setting.

# G  USE OF LARGE LANGUAGE MODELS

We employed LLMs to support the following aspects of our research:

- Ideation & Exploration: Assisting with brainstorming of methods and conducting preliminary literature searches and summarizations.
- Coding Assistance: Generating boilerplate PyTorch code, visualization scripts, and test structures. All LLM-generated code was manually reviewed and verified by the authors.
- Writing Assistance: Refining sentence structure, grammar, and clarity in the manuscript. The scientific content and all claims remain the sole work of the authors.

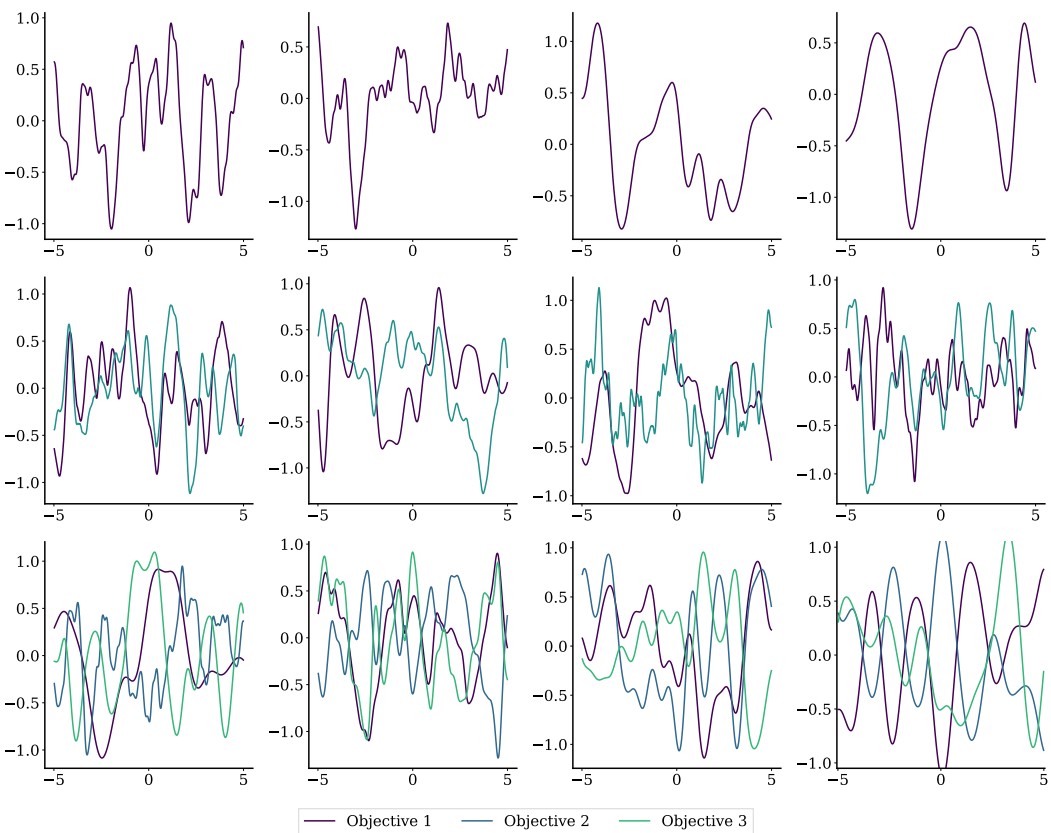

Figure S11: GP Samples used during pretraining with input dimension $d_x = 1$ and output dimension $d_y = 1, 2, 3$.

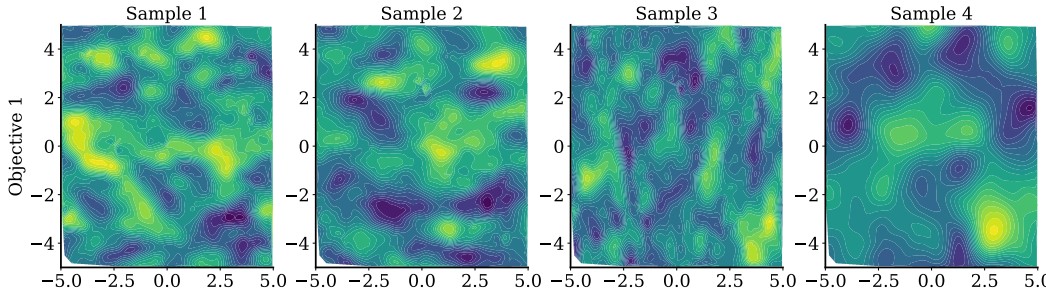

Figure S12: GP Samples used during pretraining with input dimension $d_x = 2$ and output dimension $d_y = 1$.

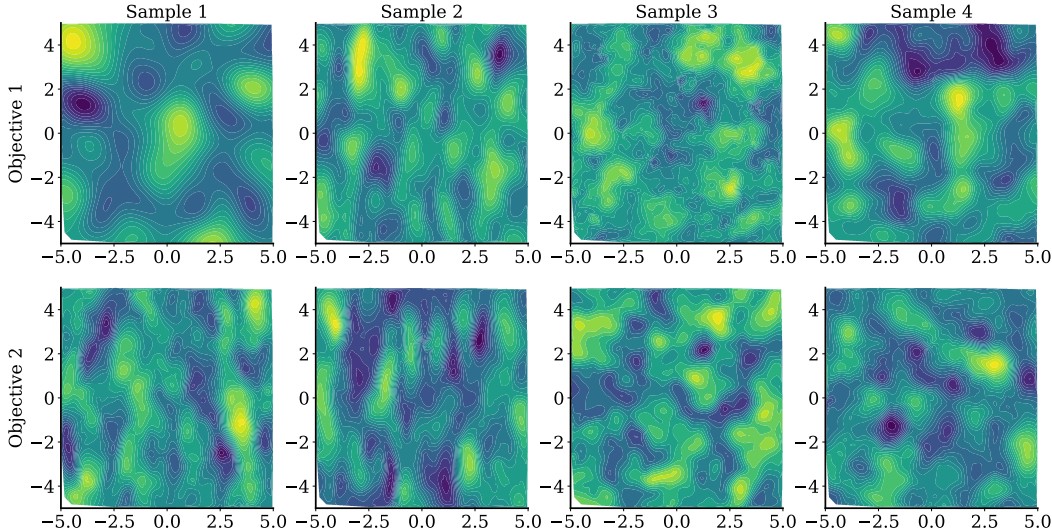

Figure S13: GP Samples used during pretraining with input dimension $d_x = 2$ and output dimension $d_y = 2$.

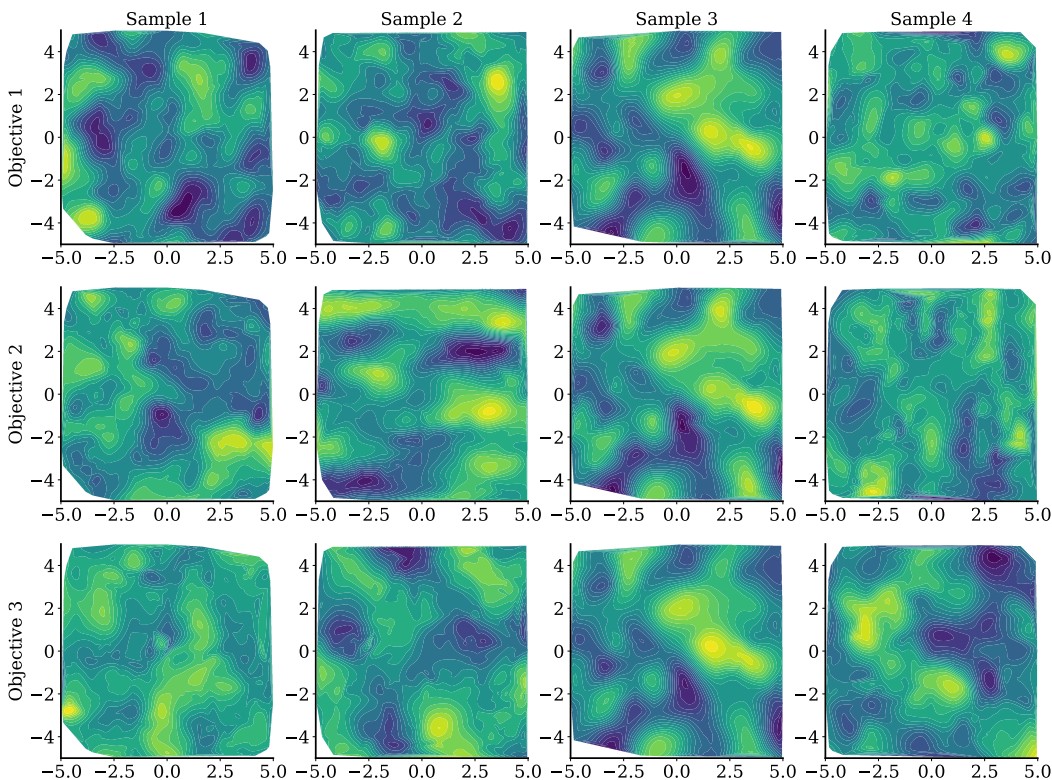

Figure S14: GP Samples used during pretraining with input dimension $d_x = 2$ and output dimension $d_y = 3$.

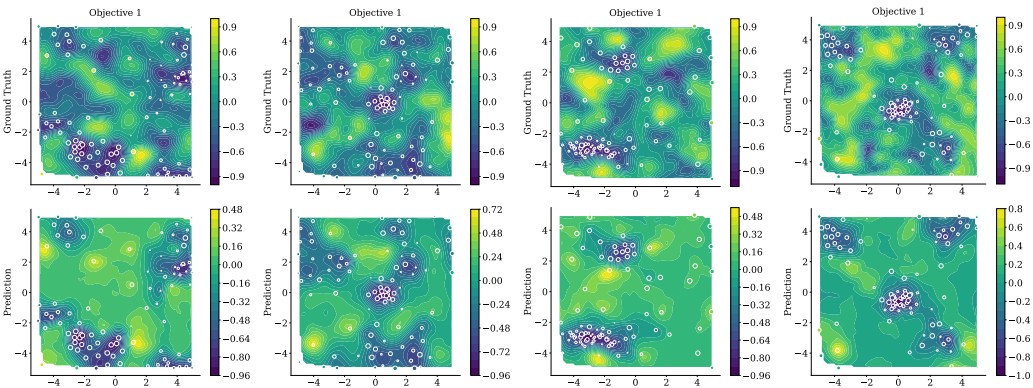

Figure S15: Inference on GP examples ($d_x = 2$, $d_y = 1$), with query points proposed over 100 optimization steps (white circle, size increasing along with the number of queries).

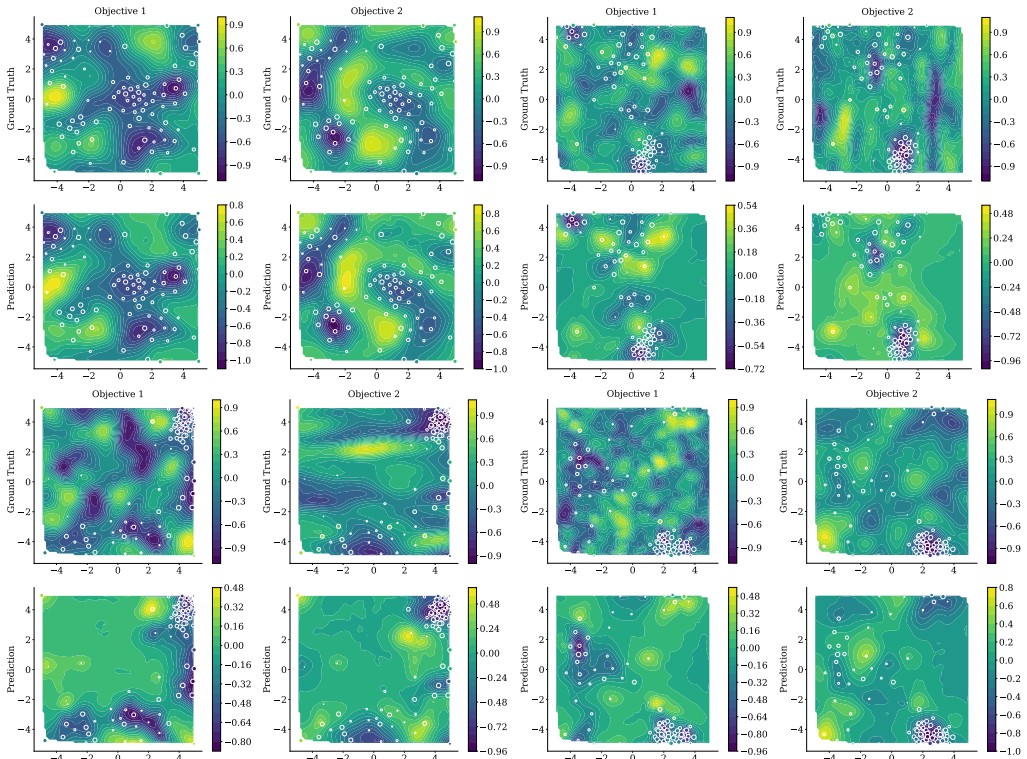

Figure S16: Inference on GP examples ($d_x = 2$, $d_y = 2$), with query points proposed over 100 optimization steps (white circles, size increasing along with the number of queries).

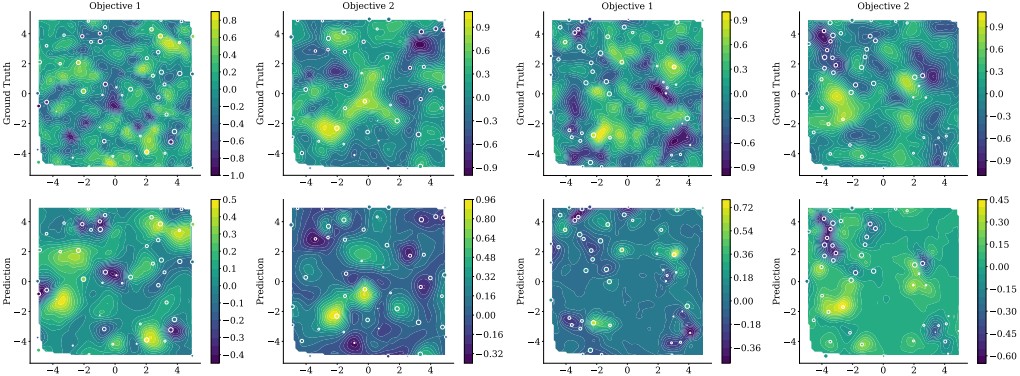

Figure S17: Mean predictions and queries on GP examples ($d_x = 2, d_y = 2$) from TAMO under the decoupled setting. Each column represents a distinct objective; queries to evaluate that objective are outlined by circles, with the sizes increasing over time to show the optimization progress. Note no queries overlap between objectives.

