# OpenReview forum: "In-Context Multi-Objective Optimization"
_ICLR.cc/2026/Conference — ICLR 2026 Poster_

### Official Review · Reviewer_mKPB · 2025-10-15

**Soundness:** 2
**Presentation:** 3
**Contribution:** 3
**Rating:** 6
**Confidence:** 3

**Summary:**

This paper proposes TAMO, a Transformer-based algorithm for black-box multi-objective optimization using a pre-trained policy. It is trained on synthetic Gaussian Process tasks with varying input and output dimensions, combining reinforcement learning and supervised losses for model-agnostic optimization. Experiments show that TAMO matches or surpasses state-of-the-art methods while being 50–1000 times faster at inference time.

**Strengths:**

1. This paper proposes a _Model-amortized Multi-objective Optimization_ algorithm that achieves significantly lower inference time compared to other MOBO methods. It could be one of the earliest works to leverage the scalability and in-context learning capability of Transformer Neural Process models to improve inference efficiency in MOBO context.

2. The paper introduces a  Dimension-agnostic embedder to the Transformer that enables dimension-agnostic learning across various MOBO tasks, making it highly generalizable to unseen tasks with varying dimensions, using a single pre-trained model. By adapting this architecture to the MOO setting, the method goes beyond a naive extension of similar approaches in single-objective Meta-BO [2] and BOFormer [1], which also implements a Transformer policy, but without dimension-agnostic learning.

**Weaknesses:**

Besides the limitations mentioned in the paper, there are some non-negligible issues:
- **Lack of statistical significance in task generalization performance** . The confidence bars of all the baselines, including TAMO, in the out-of-distribution evaluations of problems GP-DX3-DY2, GP-DX3-DY3 in Figure 4 (a) are not well-separated. Thus, Figure 4. only demonstrate limited evidence for the statistical significance of TAMO's superiority in task generalization.
- **Lack of benchmarks.** Although both synthetic and real-world benchmarks were evaluated, the input and output dimensions are relatively small (both ≤ 4), which limits the results to small-scale MOO problems. Note that the most similar and recent work BOFormer [1] includes benchmark performance with **input dimension up to 30**. Also, there is only a single real-world benchmark compared in the experiments, which is not sufficient to demonstrate the general performance of TAMO in real-world settings.

- **Lack of baseline comparisons.**

    1. The information-theoretic approaches discussed in the related work section were not included in the experimental comparisons. Other Transformer-based policies applicable to MOBO settings , such as **Optformer** [4], should also be considered for comparison, which is also compared in the paper of BOFormer [1].

    2. The performance gap between GP-based and Transformer-based models is not well studied experimentally. Given that GP and Transformer models differ substantially in scale, direct comparison is difficult. To better evaluate the usefulness of the Transformer Neural Process in terms of regret performance, variants of TAMO should be included in the ablation study, e.g., versions with RL policies using other policy parametrizations should be considered, e.g., smaller neural networks (as in [3]).
- **Necessity of the RL policy.** Although the use of Transformers is well-motivated, which possess strong in-context learning capabilities, the paper provides limited evidence supporting the need for a Transformer-based RL policy. Future experiments could include a baseline that performs GP-based MOBO (e.g., qNEHVI, qParEGO) but replaces the GP surrogates with Transformer Neural Process surrogates (pre-trained on GP priors). Comparing TAMO against such a baseline would help clarify whether using RL to learn a policy is necessary, compared with directly employing a Transformer surrogate. Moreover, this proposed baseline should also benefit from reduced inference time through forward-pass inference, thus, this comparison would also help evaluate the effectiveness of TAMO’s RL policy under similar low-inference-cost conditions

**Questions:**

See Weakness,

and

- Why is a **mixed-normal regression head** (equation 1) used for the prediction head? Would it be possible to apply a **Bar distribution (Riemann distribution) regression head** instead? [5] has shown that normal regression heads may underperform bar distribution regression heads in terms of supervised-learning efficiency in a similar setting of fitting GP priors.

- Is there a specific reason for using the **normalized hypervolume level** as the reward, rather than the **normalized hypervolume improvement** [1]? Using net improvement as reward signal is also common in RL-based BO policies [2,3] for single-objective problems. Conceptually, would using the normalized hypervolume level allow repeatedly rewarding the same query, potentially slowing down TAMO’s exploration?

- **Potentially high training time?** Although TAMO achieves significantly lower inference time than other methods, Transformer-based BO policies have been shown to require substantial training resources (as noted in [2]), even for single-objective BO. There may be a potential issue of high training cost for complex real-world problems with high input/output dimensionality. It would be helpful for the authors to discuss TAMO’s training time.

- In Section 3.2, the definition of positional tokens is confusing for me. In Line 178,
	 "These positional tokens are randomly sampled for each batch from fixed pools of learned embeddings"
	What's is the definition and the practical implementation of this "fixed pools of learned embeddings?"
- The Section 3.2 is difficult to read, I think it could be improved by providing a full model architecture illustration in Figure 2., explicitly drawing the $B_1, B_2$ blocks of layers and use arrows or attention maps to indicate which tokens can attend to which tokens in each phase.

**Minor issues**
- Figure 4. is confusing to me before I found (a), (b) in the subfigures. They look very small and are in the top of the subfigures.
- Some Figure references in the appendix are not well-compiled.


**Citations**

[1] Hung, Y. H., Lin, K.-J., Lin, Y.-H., Wang, C.-Y., Sun, C., & Hsieh, P.-C. (2025). BOFormer: Learning to solve multi-objective Bayesian optimization via non-Markovian RL. The Thirteenth International Conference on Learning Representations. [https://openreview.net/forum?id=UnCKU8pZVe](https://openreview.net/forum?id=UnCKU8pZVe)

[2] Alexandre Max Maraval, Matthieu Zimmer, Antoine Grosnit, and Haitham Bou Ammar. End-to-end
meta-bayesian optimisation with transformer neural processes. Advances in Neural Information
Processing Systems, 34, 2023.

[3] Michael Volpp, Lukas P. Fr¨ohlich, Kirsten Fischer, Andreas Doerr, Stefan Falkner, Frank Hutter,
and Christian Daniel. Meta-learning acquisition functions for transfer learning in bayesian opti-
mization. In International Conference on Learning Representations, 2020.

[4] Yutian Chen, Xiaoxi Song, Chung-Ching Lee, Zihang Wang, Ruoxi Zhang, David Dohan, Kenji
Kawakami, Greg Kochanski, Arnaud Doucet, Marc’Aurelio Ranzato, et al. Towards learning
universal hyperparameter optimizers with transformers. In Advances in Neural Information Pro-
cessing Systems, volume 35, pp. 32053–32068, 2022.

[5] Müller, S., Hollmann, N., Pineda Arango, S., Grabocka, J., & Hutter, F. (2022). Transformers can do Bayesian inference. International Conference on Learning Representations. [https://openreview.net/forum?id=KSugKcbNf9](https://openreview.net/forum?id=KSugKcbNf9)

---

> ### Author Response · Authors · 2025-11-26
> **Rebuttal 1/3**
>
> Thank you for your positive assessment of our work and the points you raised. We address your points and questions below.
>
> > 1. Lack of statistical significance in task generalization performance...Thus, Figure 4. only demonstrate limited evidence for the statistical significance of TAMO's superiority in task generalization.
>
> Thank you for raising this point. We have updated our results to include 60 independent runs. The extended evaluation confirms that the difference in performance is not statistically significant.
>
> In light of this, we have softened the wording in the main text and caption of Figure 4: we now state that TAMO attains regret broadly comparable to the strongest GP baselines and explicitly note that, even with 60 repetitions, we do not observe statistically decisive differences between methods, while TAMO retains large advantages in inference time.
>
> > 2. Lack of benchmarks.
>
> We thank the reviewer for the suggestion. To demonstrate the applicability of TAMO to a broader range of real-world scenarios, we have included a new experiment on **Hyperparameter Optimization for 3D Gaussian Splatting (HPO-3DGS)**, as evaluated in the BOFormer [1]. We evaluated TAMO against four baselines (random, qNEHVI, qNParEGO, BOFormer) across four standard scenes: Lego, Materials, Mic, and Ship. The result is shown in **Fig. S10**.
> For all scenes, qNEHVI consistently achieves the lowest simple regret. However, TAMO shows competitive performance: it achieves results comparable to the qNEHVI in the Lego and Materials scenes, and performs closely with BOFormer, while the latter requires re-training for different tasks. Most importantly, as shown in the Cumulative Inference Time plots, TAMO maintains a **decisive speed advantage**. While the inference cost of GP-based methods (including BOFormer) grows significantly with the number of queries, TAMO’s inference time is orders of magnitude lower. These results confirm that TAMO effectively generalizes to complex real-world problems.
>
> > 3. Other Transformer-based policies applicable to MOBO settings, such as Optformer, should also be considered for comparison
>
> We focused our transformer-based comparisons on BOFormer [1] rather than Optformer [2] for three key reasons: (1) Optformer is natively designed for single-objective hyperparameter optimization. Applying it to the MOBO setting requires non-trivial modifications that are not native to the original method; (2) As demonstrated in the BOFormer paper, when OptFormer was adapted for multi-objective tasks, it significantly underperforms relative to BOFormer; and (3) Since TAMO is directly compared against BOFormer, which we view as currently the best performing amortized Transformer policy for MOBO, we believe that this provides the most rigorous test of our method’s efficacy, and that including OptFormer would not provide additional insight into the state-of-the-art performance frontier.
>
> > 4. The performance gap between GP-based and Transformer-based models is not well studied experimentally. e.g., versions with RL policies using other policy parametrizations should be considered, e.g., smaller neural networks (as in [3]).
>
> We agree that understanding the effect of the policy parameterization is important. In particular, To probe the impact of model capacity within our architecture, we additionally ran a **model-size ablation** (**Fig. S8**). We compare standard TAMO (4 Transformer layers in each module: dimension-agnostic embedder, blocks $B_1$ and $B_2$ (also see Fig. S1 in the updated PDF for a detailed architecture) to a smaller TAMO version, using only 2 Transformer layers per module, trained for same number of iterations.  As shown in Fig. S8, the smaller model remains competitive but is systematically worse: on all four benchmarks it attains higher simple regret. This suggests that the additional Transformer capacity is genuinely exploited to encode richer in-context structure, rather than being an unnecessary overhead.

---

> ### Author Response · Authors · 2025-11-26
> **Rebuttal 2/3**
>
> > 5. Necessity of the RL policy. Although the use of Transformers is well-motivated, which possess strong in-context learning capabilities, the paper provides limited evidence supporting the need for a Transformer-based RL policy.
>
> We appreciate this suggestion, as it highlights the distinction between the *predictive model* (the surrogate) and the *decision-making strategy* (the policy). We believe that the RL formulation is necessary for two critical reasons: (1) **amortizing the acquisition cost**, and (2) **enabling non-myopic long-term planning**.
> We respectively point out a technical distinction regarding the reviewer’s note that a “Transformer surrogate + qNEHVI” baseline would “benefit from reduced inference time.” While replacing a Gaussian process with a Transformer neural process does speed up the *posterior prediction*, it does not remove the primary computational bottleneck of standard BO; the acquisition optimization. Standard methods like qNEHVI still require running an inner optimization loop (typically gradient ascent) over the candidate input space to maximize the acquisition value. This step is computationally expensive and does not benefit from the Transformer's speed.
>
> In contrast, TAMO is fully amortized: it generates candidate queries directly in a single forward pass, eliminating the acquisition optimization loop entirely. Consequently, the proposed baseline would be orders of magnitude slower than TAMO, making them incomparable in the context of low-latency inference. Additionally, standard acquisition functions (like qNEHVI or qParEGO) are typically myopic: they optimize for the immediate improvement of the next step. TAMO, via its RL formulation, optimizes the cumulative reward over the entire remaining horizon. Our results demonstrate that this non-myopic behavior is crucial.
>
> As shown in Figs. S6–S7 (discussed in the response to Reviewer GinK), we perform ablations on the discount factor $\gamma$ and time horizon $T$. We found that settings encouraging long-term planning ($\gamma$ close to $1$, $T=100$) consistently outperform myopic settings ($\gamma=0$, $T=1$) on difficult benchmarks. If we were to use a Transformer surrogate with a standard acquisition function (the suggested baseline), we would revert to myopic, greedy behavior. The superiority of our high-$\gamma$ agents serves as empirical evidence that the RL policy's ability to "plan" offers performance gains beyond simply having a good surrogate model.
> While a "Transformer Surrogate + qNEHVI" approach is a valid "semi-amortized" design, it fails to achieve our primary goals of (i) efficient inference latency and (ii) long-horizon non-myopic planning. We believe our existing ablations on $\gamma$ (which isolate the effect of the RL policy's planning horizon) provide sufficient evidence for the necessity of the RL approach.
>
> > 6. Why is a mixed-normal regression head (equation 1) used for the prediction head?
>
> We agree that the Riemann distribution head is a reasonable alternative. Our use of a Gaussian Mixture Model (GMM) head is a deliberate design choice motivated by the fact that GMMs are universal approximators capable of fitting arbitrary smooth densities. In contrast, the Riemann distribution relies on a discrete binning of the support with fixed buckets. Recent work on Distribution Transformers [3] explicitly compared GMM heads to the Riemann distribution used in PFNs and reported that Riemann distributions can struggle to model smooth, heavy-tailed posteriors due to this discretization, leading to worse KL/NLL especially when priors vary over a wide range. Therefore, we chose the GMM head as a simple and stable choice for our experiments.

---

> ### Author Response · Authors · 2025-11-26
> **Rebuttal 3/3**
>
> > 7. Is there a specific reason for using the normalized hypervolume level as the reward, rather than the normalized hypervolume improvement?
>
> We acknowledge that defining the reward as incremental improvement (or "utility gain") is standard practice in recent RL-based BO literature. However, we selected the **conditional normalized hypervolume level** to mitigate the issue of vanishing gradients during late-stage optimization and to ensure a consistent magnitude learning signal throughout the episode.
> To address the reviewer's conceptual concern regarding exploration, our formulation strictly precludes the accumulation of reward for redundant or non-improving queries. We employ a zero-clipping mechanism wherein the reward is nullified if the hypervolume does not strictly increase. This ensures that the agent is never rewarded for stagnation or for re-sampling existing regions of the Pareto front.
>
> Our primary motivation for using the hypervolume *level* rather than *improvement* is to stabilize training dynamics: **Improvement-based rewards** ($r_t \propto \Delta \text{HV}$) disproportionately weigh the initial steps of the optimization horizon, where large hypervolume increases are relatively trivial to achieve. Conversely, as the optimization converges, $\Delta \text{HV}$ naturally diminishes, resulting in a sparse and noisy signal during the critical fine-tuning phase. **Level-based rewards** ($r_t \propto \text{HV}_t$), when standardized across the batch, yield a **smoother and more dense reward signal**. Unlike improvement-based signals, which suffer from rapid decay, level-based rewards maintain a consistent magnitude proportional to the absolute quality of the Pareto front. This formulation encourages the agent to learn robust policies that optimize for the final solution quality, rather than policies that bias towards greedy, early-stage gains but fail to converge efficiently in the latter half of the budget.
>
> > 8. It would be helpful for the authors to discuss TAMO’s training time.
>
> It takes approximately two days to train the TAMO model evaluated in the main paper.
> 30 hours for warmup; 19.6 hours for joint training phase. Crucially, this is a one-time offline cost. Once trained, the model’s inference cost is effectively negligible (milliseconds) and offers a significant speed advantage over methods that require per-task training or extensive test-time optimization.
>
>
> > 9. In Section 3.2, the definition of positional tokens is confusing for me. In Line 178, "These positional tokens are randomly sampled for each batch from fixed pools of learned embeddings" What's is the definition and the practical implementation of this "fixed pools of learned embeddings?"
>
> Thank you for the clarifying question. The "fixed pools of learned embeddings" are two trainable parameter matrices, $P^x$ and $P^y$, for input and output positions, respectively. These matrices are of size $M_x \times H$ and $M_y \times H$, containing $M_x$ and $M_y$ distinct positional embedding vectors of dimension $H$. The pool sizes $M_x$ and $M_y$ are chosen to be “the maximum dimension”, which could be larger than the maximum dimension seen in training data. For a batch with dimensions $d_x$ and $d_y$, we randomly sample without replacement $d_x$ vectors from $P^x$ and $d_y$ vectors from $P^y$, and add them to the corresponding features and outcomes. This method decouples the positional representation capacity from the actual input dimensions in training data, allowing the model to learn a richer set of positional encodings. This prevents overfitting to a limited, fixed set of tokens and improves generalization when the model encounters novel dimensional inputs during inference.
>
> > 10. The Section 3.2 is difficult to read, I think it could be improved by providing a full model architecture illustration in Figure 2.
>
> Thanks for the suggestion. We have now included a full architecture figure along with attention masks in the updated PDF (**Fig. S1**).
>
> > 11. Figure 4. is confusing to me before I found (a), (b) in the subfigures. They look very small and are in the top of the subfigures.
>
> Thanks for pointing this out; we have updated the plots in submission to improve their readability.
>
> > 12. Some Figure references in the appendix are not well-compiled.
>
> This is now fixed, thank you for noticing.
>
> **References:**
>
> [1] Hung et al. *BOFormer: Learning to solve multi-objective Bayesian optimization via non-Markovian RL*, ICLR 2025.
>
> [2] Chen et al. *Towards learning universal hyperparameter optimizers with transformers*, NeurIPS 2022.
>
> [3] Whittle et al. *Distribution Transformers: Fast Approximate Bayesian Inference With On-The-Fly Prior Adaptation.* arXiv:2502.02463.

---

> > ### Comment · Reviewer_mKPB · 2025-11-26
> > **response**
> >
> > Thank you for addressing all my concerns and providing thoughtful discussions about the reward choice. I am happy to increase my score.

---

> > > ### Author Response · Authors · 2025-11-26
> > >
> > > Thank you for engaging with our rebuttal and raising your score. We greatly appreciate it.

---

### Official Review · Reviewer_avNh · 2025-10-31

**Soundness:** 3
**Presentation:** 3
**Contribution:** 3
**Rating:** 6
**Confidence:** 3

**Summary:**

The paper introduces TAMO, a fully amortized, task-agnostic, dimension-agnostic policy for multi-objective black-box optimization. A single transformer backbone, trained offline with (i) an in-context prediction warm-up and (ii) a policy-level RL objective that maximizes trajectory hypervolume (HV) progress, maps histories and a candidate set to the next query in one forward pass—eliminating per-task surrogate fitting and acquisition optimization. Key architectural elements include a dimension-agnostic embedder, encoder–decoder with task-specific tokens, and dual prediction/policy heads; the policy is trained with REINFORCE on HV-normalized rewards. Empirically, TAMO achieves **50×–1000×** lower proposal time than GP-based MOBO and BOFormer while delivering competitive regret on synthetic and real tasks; it also shows transfer to unseen input/output dimensionalities and decoupled observations. Code is promised upon acceptance; hyperparameters, algorithms, and synthetic pretraining data generation are documented.

**Strengths:**

* **End-to-end amortization:** Eliminates per-task surrogate fitting and acquisition optimization; single forward-pass proposals reduce decision latency by **50×–1000×**.
* **Dimension-agnostic architecture:** Embedder + task-tokens operate across varying dx, dy; supports heterogeneous pretraining and cross-dimensional transfer.
* **Non-myopic training signal:** RL objective directly optimizes trajectory HV, aligning learning with Pareto-front discovery over horizons.
* **Generalization studies:** Evidence of transfer to unseen dimensionalities and decoupled observations under fixed budget/cost accounting.
* **Methodological transparency:** Clear preliminaries, training/inference algorithms, hyperparameters; baselines (qNEHVI, qNParEGO, qHVKG, BOFormer) implemented with standard toolchains.
* **Performance profile:** Competitive or better regret across several synthetic and real tasks, with consistent runtime advantage.

**Weaknesses:**

* **OOD gaps & sensitivity:** Underperforms classic MOBO on Branin–Currin and LaserPlasma; authors attribute mismatch to pretraining length-scales—this warrants a systematic analysis.
* **Discrete candidate pool assumption:** Inference relies on a fixed candidate set; implications for high-dimensional continuous or combinatorial design spaces are acknowledged but unresolved.

**Questions:**

1. **Timing fairness & ablations:** Please report per-candidate acquisition latency (µs/ms) and GPU-to-GPU comparisons against GPyTorch/BoTorch with matched MC budgets; include breakdown of surrogate refit vs. acquisition time.
2. **Pretraining-prior sensitivity:** Provide a controlled study varying kernel families, length-scales, ARD, and output correlations to quantify transfer sensitivity and explain gaps on Branin–Currin/LaserPlasma.
3. **Continuous action spaces:** Can TAMO couple to a continuous proposal mechanism (e.g., gradient-guided refinement, learned sampler) to move beyond pool-based scoring? Any preliminary results?
4. **RL stability:** What variance-reduction techniques were used for REINFORCE (e.g., learned baseline, advantage normalization)? Any ablation on γ and λ_{rl}?
5. **Decoupled observations policy:** How are costs integrated at training time? Could explicit cost-aware reward shaping further improve decoupled performance (e.g., on Ackley–Rosenbrock)?

---

> ### Author Response · Authors · 2025-11-26
> **Rebuttal 1/3**
>
> Thank you for your positive feedback and questions. We address each of them in detail below.
>
> > 1. Timing fairness & ablations
>
> All inference times are measured as **wall-clock times** on the same GPU for all methods (GP baselines, BOFormer, TAMO), using the standard BoTorch implementations for EI, qNEHVI, qHVKG, qNParEGO. We focus on cumulative time, as it captures the practical, end-to-end cost of the optimization campaign.
>
> Figure 3 in the paper shows that the GP-based baselines have very similar runtime profiles, while TAMO (blue) is consistently 50–1000× faster at proposal time. Based on this, we provide a detailed timing breakdown for one single-objective case (Branin with EI) and one multi-objective case (Branin–Currin with qNEHVI), which are representative of the GP baselines (Figure S9)
>
> The new breakdown plots decompose cumulative inference time into surrogate refit and acquisition optimization. For Branin with EI, acquisition optimization already accounts for most of the cost by 100 iterations. For Branin–Currin with qNEHVI, acquisition optimization dominates even more strongly.
>
> These results show that, for GP-based BO, **acquisition optimization is the main bottleneck**, especially in the multi-objective setting. This is precisely the cost that TAMO removes by replacing per-step acquisition maximization (and per-task refitting) with a single forward pass through a pretrained policy, explaining the large speedups observed in Figure 3.
>
> > 2. Pretraining-prior sensitivity
>
> We fully agree that the choice of pre-training prior is central for amortized methods like TAMO. Recent work on prior-data fitted networks / TabPFN makes exactly this point: these models can be understood as approximate Bayesian predictors whose behaviour – and out-of-distribution performance in particular – is strongly shaped by the synthetic prior used for pre-training [1,2]. In that sense, designing and analysing task priors (kernel families, length-scales, ARD structure, output correlations, task diversity, …) is becoming a research area of its own, with several papers devoted entirely to understanding how prior misspecification or diversity affects in-context learners.
>
> In this work, our focus is more modest: we aim to introduce TAMO as an end-to-end Transformer policy and to show that for a fixed, fairly standard GP prior family it can match or surpass strong GP-based MOBO methods while amortizing away surrogate fitting and acquisition optimization. For clarity of comparison, we therefore deliberately kept the pre-training prior simple (stationary kernels, broad but not heavily tuned hyperpriors, independent outputs) and did not perform a full factorial study over kernel types, length-scale ranges, ARD structures, or output correlations.
>
> A truly controlled sensitivity analysis along the lines suggested by the reviewer would require retraining the high-capacity TAMO policy many times, once for each prior configuration, which is beyond what we can realistically run during the rebuttal period. We have therefore not added such an experiment in this revision, but we now explicitly flag “pre-training prior design and sensitivity” as an important avenue for follow-up work, and we see our Branin–Currin / LaserPlasma gaps as useful case studies for that future analysis.

---

> > ### Author Response · Authors · 2025-11-26
> > **Rebuttal 3/3**
> >
> > > 5. Decoupled observations policy: How are costs integrated at training time? Could explicit cost-aware reward shaping further improve decoupled performance (e.g., on Ackley–Rosenbrock)?
> >
> > In the current work, costs are **not** explicitly modeled during pretraining. We generate trajectories where each joint evaluation of all $d_y$ objectives has unit cost, and the RL reward is the (normalized) hypervolume at each step under a fixed-length horizon. The policy is thus trained to maximize trajectory hypervolume under a “one step = one full evaluation” budget, without any per-objective cost weighting.
> >
> > The decoupled experiments are then **zero-shot**: at test time, we reuse the same pretrained policy but reinterpret the horizon in terms of a cost budget $T$. A full evaluation of all $d_y$ objectives consumes cost $d_y$, while a single-objective probe costs 1, and regret is plotted against cumulative cost as described in the paper. The model itself is unchanged; it simply operates on histories where some objectives are missing or only partially observed.
> >
> > We agree that explicit **cost-aware reward shaping** is a promising way to improve decoupled performance, especially on tasks like Ackley–Rosenbrock, where objectives peak in disparate regions and single-objective probes can bias the search. Concretely, one could encode per-objective costs in the task tokens and replace the current reward with a cost-normalized signal such as $\Delta \text{HV} / \Delta \text{cost}$, allowing the policy to learn, during pretraining, when it is worth paying for expensive joint evaluations versus cheap single-objective probes.
> >
> > We believe this illustrates the versatile nature of our approach: with minimal changes to the reward and tokenization, the same architecture can naturally extend to cost-aware BO and other structured-feedback settings such as constrained BO.
> >
> > **References**
> >
> > [1] Zhang et al. *TabPFN: One Model to Rule Them All?*, arXiv:2505.20003.
> >
> > [2] Müller et al. *Position: The Future of Bayesian Prediction Is Prior-Fitted*. ICML 2025
> >
> > [3] Müller, et al. *Pfns4bo: In-context learning for Bayesian optimization*, ICML 2023.
> >
> > [4] Maraval et al. *End-to-end meta-Bayesian optimisation with transformer neural processes*, NeurIPS 2023.
> >
> > [5] Zhang et al. *PABBO: Preferential Amortized Black-Box Optimization*, ICLR 2025.

---

> > ### Comment · Reviewer_avNh · 2025-11-26
> >
> > Dear authors,
> >
> > Thank you for your response. However, I'll keep my score.

---

> > > ### Author Response · Authors · 2025-11-26
> > >
> > > Thanks for replying to our rebuttal. Your initial review raised a lot of insightful questions that we feel we engaged with deeply in our response. We noticed that your reply came within a minute of us posting the third and final part of the rebuttal. We'd kindly like to ask whether you had a chance to see all parts before replying, and if not, whether you might revisit the full response.

---

> ### Author Response · Authors · 2025-11-26
> **Rebuttal 2/3**
>
> > 3. Continuous action spaces: Can TAMO couple to a continuous proposal mechanism (e.g., gradient-guided refinement, learned sampler) to move beyond pool-based scoring?
>
> We thank the reviewer for highlighting this point. As noted in our submission, we agree that this is a limitation, and wish to emphasize that it is a shared limitation of the area rather than a specific failing of our method. The reliance on discrete candidate pools is currently the standard practice in recent work on amortized and Transformer-based black-box optimization [3-5].
> Moreover, in many practical black-box optimization problems, the design space is already handled as a pool (e.g., screened compound libraries, pre-generated molecules, candidate architectures), making the pool-based formulation well-aligned with common real-world workflows.
>
> That said, TAMO is not inherently restricted to discrete domains. Since our transformer architecture is fully differentiable, we can readily extend it to continuous domains using gradient-based refinement. Specifically, we could use the gradient of the policy's output score with respect to the input query $x$ to iteratively refine the candidate. Conceptually, a more principled solution would be to train a continuous policy head that outputs proposals directly in the continuous domain, rather than scoring a finite pool. We view this as an important direction for future work and have updated the discussion in the manuscript to reflect this: “A key challenge is scaling to higher-dimensional spaces, with promising directions including factorizing the policy across input dimensions and moving from pool-based scoring to continuous policies or generative proposal mechanisms, in the spirit of amortized design networks.”
>
> > 4. What variance-reduction techniques were used for REINFORCE? Any ablation on γ and λ_{rl}?
>
> We did not employ any variance reduction technique for REINFORCE. The effect of $\\gamma$ is analyzed in Fig S6 and discussed in our response to Reviewer GinK, in the broader context of non-myopicity. For convenience, we summarize the key point here: we vary $\\gamma \\in \\{0.25, 0.5, 0.75, 1.0\\}$ and observe that all settings eventually reach similar regret levels, but larger $\\gamma$ (more non-myopic) yields faster early regret reduction, especially on the harder tasks (Ackley-Rastrigin, Ackley-Rosenbrock). We invite the reviewer to see that answer for a broader discussion of when non-myopic training helps.
>
> Regarding $\\lambda_{rl}$: we had a notational oversight which is now fixed in the updated pdf: the weight should be associated with the auxiliary prediction loss, instead of associated with the policy loss. The impact of $\\lambda_p$ is studied in **Fig S5**, and included in a broader context necessary to understand the global impact of the prediction loss.
>
> Training proceeds in two stages. First, in a **warm-up phase**, we train only the prediction head with a supervised loss so that the shared backbone learns to model the objective functions from context/target pairs. Second, we switch to **joint training**, where the same backbone is optimized with both the RL loss (policy head) and the prediction loss. Conceptually, this follows our view that optimization is “learning to predict the optimum”: a policy can only propose good points once it has a reasonable internal model of how the objectives behave, which the warm-up provides. A pure policy-only variant would need to simultaneously discover useful representations and a good acquisition strategy from a sparse, high-variance hypervolume reward, which is considerably harder.
>
> Empirically, this is confirmed by the ablation in **Fig. S4**, where we remove the prediction term from the training loss in Eq. (6): **no warm-up and no auxiliary prediction loss**. Both models are trained on 40k datasets, but in standard TAMO the prediction head sees 16k of these datasets during warm-up. Across all four synthetic tasks, the policy-only variant (TAMO w/o prediction task) shows substantially higher simple regret and barely improves beyond its initial level, whereas standard TAMO rapidly drives regret down. This strongly suggests that the prediction warm-up is crucial for learning useful representations and, therefore, a strong optimizer.
>
> This intuition is further supported by the next ablation. During joint training, we balance the prediction and policy objectives via the weight $\\lambda_p$ in Equation 6]. The total loss is $L_{rl} + \\lambda_p L_{pred}$. **Figure S5** varies $\\lambda_p \\in \\{0.25, 0.5, 0.75, 1.0\\}$ while keeping other settings fixed.
> The resulting regret curves are very similar on all four tasks, with only a mild advantage for larger $\\lambda_p$, suggesting that once the backbone has been warmed up, performance is **fairly insensitive** to the precise trade-off between prediction and policy losses.

---

### Official Review · Reviewer_GinK · 2025-10-31

**Soundness:** 3
**Presentation:** 4
**Contribution:** 3
**Rating:** 8
**Confidence:** 3

**Summary:**

The paper targets three flaws with traditional multi-objective Bayesian optimization methods; that they require re-training from scratch for each use, the reliance on a large number of hyperparameters, and the focus on single-step gains. The authors introduce TAMO, which leverages a transformer-based optimizer trained on diverse synthetic tasks using reinforcement learning to learn multi-step strategies. Once trained, users use iterative forward passes of TAMO and a tokenized history of the steps inputted into TAMO to run the optimization process. The authors claim to match or improve the quality of state of the art methods while reducing proposal time.

**Strengths:**

The authors present a novel architecture which functions as both the acquisition function and surrogate model. The dimension-agnostic embedder is an interesting solution to generalizing over spaces with varying dimensions. TAMO is demonstrated to have equivalent or superior metrics to many other methods, and is demonstrated to run in a shorter time. The paper is overall pretty clear.

**Weaknesses:**

1. In which cases do multi-step strategies actually matter? When is myopia actually a limiting factor for Bayesian optimization? Can the authors demonstrate an ablation where multi-step strategies provide a clear advantage over traditional BO?

2. There is only one real-world benchmark. Can the authors demonstrate applicability to other real-world scenarios, such as for Gaussian splatting as done in the Boformer paper (Yu-Heng Hung, et al. 2025) or neural network hyperparameter selection as in OptFormer (Yutian Chen, et al. 2022).

3. Can the authors demonstrate clear advantages in practice to other transformer-based MOO methods, such as Optformer?

4. The authors mention that the pretraining data composition is important for generation. The authors should provide a study to experimentally demonstrate how the diversity of the pretraining data affects optimization performance. For the out-of-distribution experiments, authors should demonstrate

**Questions:**

Is the prediction task fitting to the raw data? Does the policy part of the model not converge without the prediction task? Why is this? Is there any practical application to the prediction mechanism post-training? This seems to be a central part of the architecture and is not thoroughly justified. Can the authors demonstrate an ablation for training with and without the prediction task?
Can the authors include a figure for the architecture for 3.2 (II) and (III)? The exact model architecture (where self vs cross attention is applied, where task-specific tokens are inputted) is unclear simply due to wordiness and a simple diagram could very clearly explain it.

---

> ### Author Response · Authors · 2025-11-26
> **Rebuttal 1/3**
>
> We sincerely thank you for your positive assessment of our paper. We appreciate your thoughtful suggestions and address your questions below.
>
> > 1. In which cases do multi-step strategies actually matter? When is myopia actually a limiting factor for Bayesian optimization? Can the authors demonstrate an ablation where multi-step strategies provide a clear advantage over traditional BO?
>
> Conceptually, multi-step strategies are most beneficial when the evaluation budget is small relative to the problem difficulty, so each query must be chosen with its downstream impact in mind. Prior work on non-myopic BO shows that explicit multi-step lookahead can significantly outperform one-step EI and related baselines on such “hard’’ functions (multi-modal, deceptive landscapes, narrow basins), while remaining computationally tractable. For instance, [1] reports consistent gains of 2-4-step lookahead over Expected Improvement.
>
> In TAMO, non-myopic behavior is controlled by the RL discount factor $\gamma$ and the pretraining horizon $T$ in the RL loss (Equation 4). Figure S6 varies $\gamma \in \{0.25, 0.5, 0.75, 1\} $: all settings eventually reach similar regret levels, but larger $\gamma$ (more non-myopic) accelerates early regret reduction, especially on harder tasks such as Ackley–Rastrigin and Ackley–Rosenbrock.
> Figure S7 compares a __myopic__ TAMO variant pretrained to maximize the reward of the single next step to our standard __multi-step__ TAMO pretrained with $T=100$.
> In the myopic case, each pretraining episode starts from a larger randomly sampled context set and the policy is trained to optimize only the next step; in the multi-step case, episodes start from a single initial point and the policy is trained over trajectories of length up to 100.
>  For both single- and multi-objective scenarios, the non-myopic variant clearly achieves overall lower regret and/or faster convergence. Together, these results indicate that multi-step training provides a tangible advantage over myopic policies, and that our RL formulation effectively exploits this non-myopic signal.
>
> > 2. There is only one real-world benchmark.
>
> We thank the reviewer for the suggestion. To demonstrate the applicability of TAMO to a broader range of real-world scenarios, we have included a new experiment on __Hyperparameter Optimization for 3D Gaussian Splatting (HPO-3DGS)__, as evaluated in the BOFormer [2]. We evaluated TAMO against four baselines (random, qNEHVI, qNParEGO, BOFormer) across four standard scenes: Lego, Materials, Mic, and Ship. The result is shown in Figure S10
> For all scenes, qNEHVI consistently achieves the lowest simple regret. However, TAMO shows competitive performance: it achieves results comparable to the qNEHVI in the Lego and Materials scenes, and performs closely with BOFormer, while the latter requires re-training for different tasks. Most importantly, as shown in the Cumulative Inference Time plots, TAMO maintains a __decisive speed advantage__. While the inference cost of GP-based methods (including BOFormer) grows significantly with the number of queries, TAMO’s inference time is orders of magnitude lower. These results confirm that TAMO effectively generalizes to complex real-world problems.

---

> ### Author Response · Authors · 2025-11-26
> **Rebuttal 2/3**
>
> > 3. Can the authors demonstrate clear advantages in practice to other transformer-based MOO methods, such as Optformer?
>
> To the best of our knowledge, there are two main lines that are closely related to our work:
>
> - BOFormer [2], to our knowledge, is the only published Transformer architecture that directly targets multi-objective Bayesian optimization. However, BOFormer is still built on a __GP surrogate__. Therefore, it is not a fully end-to-end policy. Moreover, the input representation and training setup assume a __fixed number of objectives__, so changing the output dimensionality requires retraining the model. In contrast, TAMO learns a fully amortized policy. The architecture is output-dimension agnostic, so the same policy can be applied to problems with different numbers of objectives without retraining. Empirically, we already compare against BOFormer in our experiments: our policy shows overall better performance than BOFormer while offering substantial speedups.
>
> - OptFormer [3] is another Transformer-based model, however, it differs fundamentally from TAMO in scope and architecture. OptFormer was designed and trained for __single-objective__ HPO. While theoretically extensible to MOO, it lacks the native components to represent Pareto sets and hypervolume. Also, OptFormer treats optimization as a "text-to-text" language modeling problem. This tokenization scheme is not designed for numerical embeddings and scales poorly to long trajectories due to each observation in the trajectory being represented by a list of text-based tokens describing the feature values for that particular observation. TAMO uses __embeddings designed for general numerical tasks__, allowing it to efficiently process long horizons and continuous numerical spaces, by only requiring a single token per new observation in the trajectory. In summary, OptFormer is tailored specifically for HPO tasks with rich textual metadata. TAMO is designed as a general-purpose optimizer, capable of functioning on purely numerical problems where no semantic metadata exists.
>
> > 4. The authors should provide a study to experimentally demonstrate how the diversity of the pretraining data affects optimization performance.
>
> We fully agree that the choice of pre-training prior is central for amortized methods like TAMO. Recent work on prior-data fitted networks / TabPFN makes exactly this point: these models can be understood as approximate Bayesian predictors whose behaviour – and out-of-distribution performance in particular – is strongly shaped by the synthetic prior used for pre-training [4,5]. In that sense, designing and analysing task priors (kernel families, length-scales, ARD structure, output correlations, task diversity, …) is becoming a research area of its own, with several papers devoted entirely to understanding how prior misspecification or diversity affects in-context learners.
>
> In this work, our focus is more modest: we aim to introduce TAMO as an end-to-end Transformer policy and to show that for a fixed, fairly standard GP prior family it can match or surpass strong GP-based MOBO methods while amortizing away surrogate fitting and acquisition optimization. For clarity of comparison, we therefore deliberately kept the pre-training prior simple (stationary kernels, broad but not heavily tuned hyperpriors, independent outputs) and did not perform a full factorial study over kernel types, length-scale ranges, ARD structures, or output correlations.
>
> A truly controlled sensitivity analysis along the lines suggested by the reviewer would require retraining the high-capacity TAMO policy many times, once for each prior configuration, which is beyond what we can realistically run during the rebuttal period. We have therefore not added such an experiment in this revision, but we now explicitly flag “pre-training prior design and sensitivity” as an important avenue for follow-up work, and we see our Branin–Currin / LaserPlasma gaps as useful case studies for that future analysis.

---

> ### Author Response · Authors · 2025-11-26
> **Rebuttal 3/3**
>
> > 5. Is the prediction task fitting to the raw data?
>
> Before being processed by the model, we linearly rescale the input space and normalize the objective values for each task. This normalization trick ensures that the model is invariant to the scale of the objective functions, allowing the shared backbone to generalize effectively across heterogeneous problem settings.
>
> > 6. Can the authors demonstrate an ablation for training with and without the prediction task?
>
> This question was also raised by Reviewer eMcs. For convenience, we reproduce below the relevant part of our response and refer the reviewer to that answer for further details on the ablation.
> Training proceeds in two stages: (1) a warmup phase where we train only the prediction head with a supervised prediction loss, followed by (2) jointly training the prediction and policy heads with an RL and prediction loss respectively (Eq. (6)).  In the warmup, the shared backbone learns to model objective functions solely from context/target pairs. This aligns with the view that optimization is “learning to predict the optimum”: a policy can only efficiently propose candidates once it possesses a reasonable internal model of the objective landscape, which the warm-up provides. A pure policy-only variant faces the significantly harder challenge of simultaneously learning representations and acquisition strategies solely from a sparse, high-variance hypervolume reward.
>
> Empirically, this is confirmed by the ablation in __Fig. S4__ in the updated PDF, where we compare standard TAMO against a variant with __no warm-up and no auxiliary prediction loss__. Both models are trained on 40k datasets (with standard TAMO using 16k of these datasets during warmup). Across all four tasks, the policy-only variant (TAMO w/o prediction task) exhibits substantially higher simple regret and fails to improve significantly beyond its random initialization, whereas standard TAMO rapidly reduces in simple regret. This confirms that the prediction warm-up is crucial for learning useful representations.
>
> This is further supported by the next ablation. During joint training, we balance the prediction and policy objectives via the weight $\lambda_p$ in Equation 6 __[This weight used to be associated with the policy loss, which was a notational oversight, as interpreting the prediction task as an auxiliary term that may get re-weighted is much more natural; we have now fixed the submission accordingly]__. The total loss is $L_{RL} + \lambda_p L_{pred}$. Figure S5 varies $\lambda_p \in \{0.25, 0.5, 0.75, 1.0\}$ while keeping other settings fixed.
> The resulting regret curves are very similar on all four tasks, with only a mild advantage for larger $\lambda_p$, suggesting that once the backbone has been warmed up, performance is __fairly insensitive__ to the precise trade-off between prediction and policy losses.
>
>
> > 7. Is there any practical application to the prediction mechanism post-training?
>
> That’s a good question. We believe that the prediction head is not merely helping stabilize the policy learning but could serve crucial practical roles during deployment. For example, while the policy head outputs a scalar utility, the prediction head provides the estimated mean and variance of the objective landscape. This allows practitioners to __visualize the landscape__ and understand why a candidate was selected (e.g., discerning between exploration of high-uncertainty regions vs. exploitation of high-mean regions). Besides, the predictive uncertainty estimates could serve as a metric for __convergence diagnosis__, allowing users to terminate optimization early if the model becomes confident that no further significant gains are likely, saving experimental costs.
>
> > 8. Can the authors include a figure for the architecture for 3.2 (II) and (III)?
>
> Thanks for the suggestion. We have now included a full architecture figure along with attention masks in the updated PDF (Fig.S1).
>
> **References**
>
> [1] Jiang et al. *Efficient Nonmyopic Bayesian Optimization via One-Shot Multi-Step Trees*, Neurips 2020.
>
> [2] Hung et al. *BOFormer: Learning to solve multi-objective Bayesian optimization via non-Markovian RL*, ICLR 2025.
>
> [3] Chen et al. *Towards learning universal hyperparameter optimizers with transformers*, NeurIPS 2022.
>
> [4] Zhang et al. *TabPFN: One Model to Rule Them All?*, arXiv:2505.20003.
>
> [5] Müller et al. *Position: The Future of Bayesian Prediction Is Prior-Fitted*, ICML 2025.

---

### Official Review · Reviewer_eMcs · 2025-11-09

**Soundness:** 3
**Presentation:** 3
**Contribution:** 3
**Rating:** 4
**Confidence:** 2

**Summary:**

This paper introduces TAMO, a novel amortized policy for multi-objective black-box
optimization, aiming to address the high computational cost and task-specific nature of GP-based
methods. The core idea is to pre-train a single, dimension-agnostic Transformer on a diverse
corpus of synthetic tasks. This pre-trained model can then be deployed on new, unseen problems,
replacing the slow, iterative refitting of a surrogate model with a single, fast forward pass to
propose the next query. The model is trained using a combination of a prediction loss and a
non-myopic RL objective based on cumulative hypervolume improvement .

**Strengths:**

● Paper is very well written and easy to understand
● By replacing the iterative GP refitting and acquisition optimization process with a single
neural network forward pass, the method reduces inference latency by 50-1000x.
● The proposed dimension-agnostic embedder is a clever architectural contribution . It
allows a single Transformer backbone to be pre-trained on and deployed to problems of
varying input and output dimensions

**Weaknesses:**

● The framework relies on a pre-defined discrete candidate pool from which the policy
head selects the next query. This is a significant limitation as it makes the approach
unusable for true continuous-domain optimization or generative tasks (like de novo drug
design).
● The "task-agnostic" claim is weakened by the model's sensitivity to the pre-training data.
The authors hypothesize that the poor performance on BraninCurrin stems from not
seeing those objective properties in the synthetic pre-training corpus. The model isn't
truly "agnostic" but is rather "multi-task" for a specific family of synthetic GP-based
tasks.

**Questions:**

● The model has two heads (Prediction and Policy) and is trained with a joint loss.
However, the contribution of the auxiliary prediction task L(p) is not ablated. How much
does the "warm-up" and joint training contribute to the final policy versus simply training
the policy head alone?

---

> ### Author Response · Authors · 2025-11-26
> **Rebuttal 1/2**
>
> We thank the reviewer for the thoughtful review and appreciate their positive comments: that the paper is well-written, the method is efficient, and that the architecture is clever and a nice contribution. We provide responses to their comments below.
>
> > 1. The framework relies on a pre-defined discrete candidate pool.
>
> We thank the reviewer for highlighting this point. As noted in our submission, we agree that this is a limitation, and wish to emphasize that it is a shared limitation of the area rather than a specific failing of our method. The reliance on discrete candidate pools is currently the standard practice in recent work on amortized and Transformer-based black-box optimization [1-3].
> Moreover, in many practical black-box optimization problems, the design space is already handled as a pool (e.g., screened compound libraries, pre-generated molecules, candidate architectures), making the pool-based formulation well-aligned with common real-world workflows.
>
> That said, TAMO is not inherently restricted to discrete domains. Since our transformer architecture is fully differentiable, we can readily extend it to continuous domains using gradient-based refinement. Specifically, we could use the gradient of the policy's output score with respect to the input query $x$ to iteratively refine the candidate. Conceptually, a more principled solution would be to train a continuous policy head that outputs proposals directly in the continuous domain, rather than scoring a finite pool. We view this as an important direction for future work and have updated the discussion in the manuscript to reflect this:
> “A key challenge is scaling to higher-dimensional spaces, with promising directions including factorizing the policy across input dimensions and moving from pool-based scoring to continuous policies or generative proposal mechanisms, in the spirit of amortized design networks.”
>
> > 2. The "task-agnostic" claim is weakened by the model's sensitivity to the pre-training data.
>
> We clarify that in our work, “task-agnostic” refers to the architecture, and not the prior.  It denotes that a single pretrained policy can be applied, without retraining or architectural changes, across heterogeneous optimization tasks with varying input or objective dimensionalities. In contrast, classical GP or  BOFormer pipelines require refitting surrogates or re-optimizing acquisition functions for each new task.
>
> Regarding the sensitivity to the pretraining data: We agree with the reviewer that performance depends on the pretraining corpus. This is consistent with the prior-fitted network view of learning: our model approximates some functional of the synthetic prior used during training. Therefore, the dependence on the pretraining corpus is simply the amortized equivalent of the dependence on the prior in classical Bayesian inference. The lower performance on certain functions such as BraninCurrin is not a failure of the “task-agnostic” architecture, but a case of prior misspecification. As a result, the composition of the synthetic pre-training corpus is itself an important and active area of research. This can be addressed by broadening the kernel families in the synthetic corpus (e.g., including non-stationary or anisotropic kernels) without changing the TAMO architecture. __We have updated the draft to elaborate on this point at the discussion level. More precisely, we added this sentence:__
>
> ‘’Lastly, as mentioned above, further work will investigate how the composition of the synthetic pre-training corpus influences downstream performance, an important direction for improving robustness and out-of-distribution behavior of amortized BO policies.’’

---

> ### Author Response · Authors · 2025-11-26
> **Rebuttal 2/2**
>
> > 3. How much does the "warm-up" and joint training contribute to the final policy versus simply training the policy head alone?
>
> Training proceeds in two stages: (1) a warmup phase where we train *only* the prediction head with a supervised prediction loss, followed by (2) jointly training the prediction and policy heads with an RL and prediction loss respectively (Eq. (6)).  In the warmup, the shared backbone learns to model objective functions solely from context/target pairs. This aligns with the view that optimization is “learning to predict the optimum”: a policy can only efficiently propose candidates once it possesses a reasonable internal model of the objective landscape, which the warm-up provides. A pure policy-only variant faces the significantly harder challenge of simultaneously learning representations and acquisition strategies solely from a sparse, high-variance hypervolume reward.
>
> Empirically, this is confirmed by the ablation in **Fig. S4** in the updated PDF, where we compare standard TAMO against a variant with **no warm-up and no auxiliary prediction loss.** Both models are trained on 40k datasets (with standard TAMO using 16k of these datasets during warmup), of input dimension $d_x = 2$ and output dimension $d_y=2$. Across all four tasks, the policy-only variant (TAMO w/o prediction task) exhibits substantially higher simple regret and fails to improve significantly beyond its random initialization, whereas standard TAMO rapidly reduces in simple regret. This confirms that the prediction warm-up is crucial for learning useful representations.
>
> This is further supported by the next ablation. During joint training, we balance the prediction and policy objectives via the weight $\\lambda_p$ in Equation 6 **\[This weight used to be associated with the policy loss, which was a notational oversight, as interpreting the prediction task as an auxiliary term that may get re-weighted is much more natural; we have now fixed the submission accordingly**\]. The total loss is $L_{RL} \+ \\lambda_p L_{pred}$. **Figure S5** varies $\\lambda\_p \\in \\{0.25, 0.5, 0.75, 1.0\\}$ while keeping other settings fixed.
> The resulting regret curves are very similar on all four tasks, with only a mild advantage for larger $\\lambda_p$, suggesting that once the backbone has been warmed up, performance is **fairly insensitive** to the precise trade-off between prediction and policy losses.
>
> **References**
> \[1\] Müller, et al. *Pfns4bo: In-context learning for Bayesian optimization*, ICML 2023.
> \[2\] Maraval et al. *End-to-end meta-Bayesian optimisation with transformer neural processes*, NeurIPS 2023.
> \[3\] Zhang et al. *PABBO: Preferential Amortized Black-Box Optimization*, ICLR 2025.

---

### Author Response · Authors · 2025-11-26
**Global comment**

We thank all four reviewers for their careful reading and constructive feedback. Across the reviews, they highlighted:

- Originality of learning a single pretrained policy for entire classes of MOBO problems
- Generality and novelty of the dimension-agnostic architecture in MOO
- Strong empirical results for 50x-1000x lower proposal times relative to baselines
- Clarity and transparency of the presentation

The reviewers raised comments regarding the following points:
- Scope of empirical validation
- The “task-agnosticity” of the architecture
- Architectural and training details

We have now updated the submission PDF with additional information, including new experimental results in response to the reviewer’s comments.

We are very grateful to the reviewers for their thorough and constructive engagement. Their feedback has been invaluable and has helped us significantly improve the clarity and empirical validation of our work.

---

### Author Response · Authors · 2025-12-03
**Final summary comment**

We thank the reviewers and ACs once more for their time and feedback throughout the process. Since the original submission, we have added the following new analyses and experiments to the updated PDF.

- Effect of prediction warm-up and the prediction-term weight $\lambda_p$ in the policy training loss (Figs S4 and S5)
- Ablation study on the discount factor $\gamma$ in the RL objective (Fig S6)
- Comparison between the standard multi-step and myopic TAMO variants (Fig S7)
- Effect of model size on optimization performance (Fig S8)
- Timing breakdown for GP baselines (surrogate refit vs. acquisition optimization) (Fig S9)
- Evaluation on real-world HPO-3DGS hyperparameter optimization tasks (Fig S10)

- **Effect of pre-training dataset composition (pretraining-prior sensitivity) (Fig S11)**

On the last point, which was requested by several reviewers, we now provide an explicit pretraining-prior sensitivity study. We train two extra TAMO variants with modified synthetic priors: (i) a __small-lengthscale__ GP prior, where lengthscales are truncated from $[0.1, 2.0]$ to $[0.1, 0.5]$, and (ii) a __quadratic-bowl__ prior, where for each of the $1 \le m \le d_y$ objectives we draw a GP, sample an "optimum location" $x^{\star(m)}$ uniformly in the design space, and add a term $\lVert x - x^{\star(m)} \rVert^2$ to the $m$-th objective.

For the small-lengthscale prior, we generally observe slightly worse performance than the original TAMO, except on Ackley–Rastrigin where both objectives are naturally well modelled by short-scale structure and performance improves. For the quadratic-bowl prior, performance improves on the Ackley-based tasks but degrades on Branin–Currin, which is consistent with Branin having multiple optima: biasing pretraining toward single-optimum landscapes appears to hurt transfer to multimodal problems. Overall, these results confirm that the pretraining prior shapes downstream behaviour in meaningful ways, but they also show that reasonable changes to the prior lead to moderate shifts rather than catastrophic failure.

Post rebuttal, reviewer mKPB increased their score from 6 to 8, explicitly noting that the new results and clarifications addressed their main concerns; this change was made before the recent OpenReview leakage and was later reverted to 6 by the system. Reviewer avNh acknowledged our answers but chose to keep their score unchanged; their follow-up reply arrived less than a minute after our detailed rebuttal, leaving us unsure to what extent the new material was taken into account.

We respectfully ask the ACs to consider the full exchange in their final assessment, including new experiments, and post-rebuttal score update by mKPB.

---

### Meta-Review · Area_Chair_TCHi · 2026-01-07

**Summary:**

This paper proposes TAMO, a fully amortized transformer policy for multi-objective black-box optimization that replaces per-task surrogate refitting and acquisition optimization with a single forward-pass proposal step. The core technical idea is a dimension-agnostic architecture (handling varying input and objective dimensions) combined with a two-stage training scheme (prediction warm-up + RL to maximize cumulative hypervolume over trajectories) to learn non-myopic optimization behavior. Reviewers broadly agreed the approach is novel and timely, and that the main practical benefit is a very large reduction in proposal latency (reported as 50–1000×) while maintaining competitive Pareto quality under small evaluation budgets. The main concerns were (i) sensitivity to the pretraining prior and OOD gaps on specific tasks (e.g., Branin–Currin, LaserPlasma), (ii) reliance on a discrete candidate pool rather than true continuous optimization, (iii) scope of empirical validation (more real-world tasks and broader dimensions), and (iv) clarity and necessity of the prediction warm-up / training details, plus timing fairness details for GP baselines.

I recommend acceptance: This is a strong and timely contribution: it introduces a genuinely end-to-end amortized MOBO policy that appears to remove the dominant per-step bottleneck (acquisition optimization) while remaining competitive in Pareto quality under strict budgets. The authors’ added ablations and experiments directly address multiple review-critical questions (warm-up necessity, non-myopia, timing breakdown, model size, additional real-world benchmark, and explicit pretraining-prior sensitivity). While robustness to prior misspecification and the discrete pool assumption remain important limitations, they are clearly articulated, partially stress-tested, and do not undermine the main contribution in the settings evaluated.

**Reviewer Concerns:**

Concerns addressed by the rebuttal:

* Prediction warm-up necessity and the role of the auxiliary prediction loss:
Authors added an explicit ablation showing policy-only training (no warm-up and no prediction loss) fails to learn effectively, while standard TAMO improves substantially (Fig. S4), plus sensitivity to the prediction-loss weight (Fig. S5).
* Non-myopic vs myopic behavior:
Authors added an ablation on discount factor gamma (Fig. S6) and a direct comparison between multi-step TAMO and a myopic variant (Fig. S7), supporting the claim that multi-step training yields faster/better convergence on harder tasks.
* Timing fairness and where GP time is spent:
Authors added a timing breakdown separating surrogate refit vs acquisition optimization (Fig. S9), clarifying that acquisition optimization dominates and is precisely what TAMO amortizes away.
* Empirical scope:
Authors added a new real-world benchmark (HPO for 3D Gaussian Splatting) with comparisons including BOFormer and GP baselines (Fig. S10), and added a model-size ablation (Fig. S8).
* Pretraining-prior sensitivity:
Authors added an explicit prior sensitivity study with two modified synthetic priors (small-lengthscale GP and a quadratic-bowl prior) and reported task-dependent, non-catastrophic shifts (Fig. S11).
* Architecture clarity:
Authors added a full architecture figure with attention masks (Fig. S1), addressing requests for a clearer depiction.

Concerns partially addressed or still outstanding:
* OOD gaps and robustness:
The added sensitivity study is a meaningful step, but it also reinforces that downstream performance depends materially on the pretraining prior. Some gaps relative to strong GP-based MOBO persist on certain tasks; the work frames this as prior misspecification rather than an architectural limitation, which is reasonable but leaves robustness as an open limitation.
* Discrete candidate pool limitation:
Authors acknowledge this and discuss possible continuous extensions (e.g., gradient-guided refinement or continuous policy heads), but these remain future work; as-is, applicability is strongest for pool-based settings.
* RL training practicality:
Authors report a nontrivial one-time training cost (on the order of days) and emphasize amortization. This is acceptable for a “foundation optimizer” framing, but it remains a real tradeoff versus per-task BO in settings where amortization is not reused many times.

**Reviewer Scores:**

Reviewer GinK (score 8): Likely unchanged (8). The reviewer’s main asks were multi-step vs myopic evidence, more real-world validation, comparisons to transformer-based methods, pretraining diversity, and architecture clarity. The authors added direct multi-step ablations, an additional real-world benchmark, and an architecture figure; the remaining limitations are acknowledged rather than fully resolved, which is consistent with maintaining a strong accept.

Reviewer avNh (score 6): Likely unchanged (6). The reviewer’s main issues were prior sensitivity/OOD gaps and discrete pool assumptions; the authors provided timing breakdowns and added some prior sensitivity experiments, but did not fully resolve robustness concerns (and could not feasibly do a full factorial prior study). This aligns with keeping a cautious marginal accept.

Reviewer mKPB (score 6; indicated they would increase post-rebuttal): Likely increases to 7 if full discussion had occurred. The authors directly addressed several concrete critiques (significance wording softened with 60 runs; added more benchmarks; clarified reward choice; discussed training time; improved architecture clarity and figure issues; added model-size ablation). The reviewer explicitly stated they were happy to increase their score after the rebuttal.

Reviewer eMcs (score 4): Likely increases to 5. The reviewer’s main question was an ablation on the prediction warm-up / auxiliary task, which the authors added (Fig. S4/S5). The other concerns (candidate pool limitation and “task-agnostic” meaning) are acknowledged and clarified, though not fully eliminated; a modest increase seems plausible.

---

### Decision · Program_Chairs · 2026-01-26

Accept (Poster)